

# Monthly Gridded Data Product of Northern Wetland Methane Emissions Based on Upscaling Eddy Covariance Observations

Olli Peltola[1], Timo Vesala[2,3], Yao Gao[1], Olle Räty[4], Pavel Alekseychik[5], Mika Aurela[1], Bogdan Chojnicki[6], Ankur R. Desai[7], Albertus J. Dolman[8], Eugenie S. Euskirchen[9], Thomas Friborg[10], Mathias Göckede[11], Manuel Helbig[12], Elyn Humphreys[13], Robert B. Jackson[14], Georg Jocher[15*], Fortunat Joos[16], Janina Klatt[17], Sara H. Knox[18], Lars Kutzbach[19], Sebastian Lienert[16], Annalea Lohila[1,2], Ivan Mammarella[2], Daniel F. Nadeau[20], Mats B. Nilsson[15], Walter C. Oechel[21,22], Matthias Peichl[15], Thomas Pypker[23], William Quinton[24], Janne Rinne[25], Torsten Sachs[26], Mateusz Samson[6], Hans Peter Schmid[17], Oliver Sonnentag[27], Christian Wille[26], Donatella Zona[21,28], Tuula Aalto[1]

[1] Climate Research Programme, Finnish Meteorological Institute, P.O. Box 503, 00101 Helsinki, Finland

[2] Institute for Atmosphere and Earth System Research/Physics, PO Box 68, Faculty of Science, FI-00014, University of Helsinki, Finland

[3] Institute for Atmospheric and Earth System Research/Forest Sciences, PO Box 27, Faculty of Agriculture and Forestry, FI-00014, University of Helsinki, Finland

[4] Meteorological Research, Finnish Meteorological Institute, P.O. Box 503, 00101 Helsinki, Finland

[5] Natural Resources Institute Finland (LUKE), FI-00790 Helsinki, Finland

[6] Department of Meteorology, Faculty of Environmental Engineering and Spatial Management, Poznan University of Life Sciences, 60-649 Poznan, Poland

[7] Department of Atmospheric and Oceanic Sciences, University of Wisconsin-Madison, 1225 W Dayton St, Madison, Wisconsin 53706 USA, 608-265-9201, desai@aos.wisc.edu

[8] Department of Earth Sciences, Faculty of Sciences, Vrije Universiteit Amsterdam, Boelelaan 1085, 1081 HV Amsterdam, the Netherlands

[9] University of Alaska Fairbanks, Institute of Arctic Biology, 2140 Koyukuk Dr., Fairbanks, AK 99775

[10] Department of Geosciences and Natural Resource Management, University of Copenhagen , Denmark

[11] Max Planck Institute for Biogeochemistry, Jena, Germany

[12] School of Geography and Earth Sciences, McMaster University, Hamilton, ON, Canada

[13] Department of Geography & Environmental Studies, Carleton University, Ottawa, ON, Canada

[14] Department of Earth System Science, Woods Institute for the Environment, and Precourt Institute for Energy, Stanford University, Stanford, CA 94305, USA

[15] Department of Forest Ecology and Management, Swedish University of Agricultural Sciences, Umeå, Sweden

[16] Climate and Environmental Physics, Physics Institute and Oeschger Centre for Climate Change Research, University of Bern, Bern, Switzerland

[17] Institute of Meteorology and Climatology – Atmospheric Environmental Research (IMK-IFU), Karlsruhe Institute of Technology (KIT), Kreuzeckbahnstrasse 19, 82467 Garmisch-Partenkirchen, Germany

[18] Department of Geography, The University of British Columbia, Vancouver, Canada

[19] Institute of Soil Science, Center for Earth System Research and Sustainability, Universität Hamburg, Hamburg 20146, Germany

[20] Department of Civil and Water Engineering, Université Laval, Quebec City, Canada

[21] Global Change Research Group, Dept. Biology, San Diego State University, San Diego, CA 92182, USA

[22] Department of Geography, College of Life and Environmental Sciences, University of Exeter, Exeter, EX4 4RJ, UK

[23] Department of Natural Resource Sciences, Thompson Rivers University, Kamloops, BC, Canada

[24] Cold Regions Research Centre, Wilfrid Laurier University, Waterloo, ON, Canada

[25] Department of Physical Geography and Ecosystem Science, Lund University, Lund, Sweden

[26] GFZ German Research Centre for Geosciences, Potsdam, Germany



[27] Département de géographie, Université de Montréal, Montréal, QC H2V 3W8, Canada
[28] Department of Animal and Plant Sciences, University of Sheffield, Western Bank, Sheffield, S10 2TN, United Kingdom
*Georg Jocher now at: Department of Matter and Energy Fluxes, Global Change Research Institute, Czech Academy of Sciences, Bělidla 986/4a, 603 00 Brno, the Czech Republic

*Correspondence to*: Olli Peltola (olli.peltola@fmi.fi)

**Abstract.** Natural wetlands constitute the largest and most uncertain source of methane ($CH_4$) to the atmosphere and a large fraction of them are in the northern latitudes. These emissions are typically estimated using process (bottom-up) or inversion (top-down) models, yet the two are not independent of each other since the top-down estimates rely on the a priori estimation of these emissions coming from the process models. Hence, independent validation data of the large-scale emissions would be

needed.

Here we utilize random forest (RF) machine learning technique to upscale $CH_4$ eddy covariance flux measurements from 25 sites to estimate $CH_4$ wetland emissions from the northern latitudes (north of 45 °N) during years 2013 and 2014. The predictive performance of the RF model is evaluated using the leave-one-site-out cross-validation scheme and the performance (Nash-Sutcliffe model efficiency = 0.47) is comparable to previous studies upscaling net ecosystem exchange of carbon dioxide or

studies where process models are compared against site-level $CH_4$ emission data. Three wetland maps are utilized in the upscaling and the annual emissions for the northern wetlands yield 31.7 (22.3-41.2, 95 % confidence interval), 30.6 (21.4-39.9) or 37.6 (25.9-49.5) $Tg(CH_4)$ $yr^{-1}$, depending on the map used. To evaluate the uncertainties of the upscaled product it is also compared against two process models (LPX-Bern and WetCHARTs) and methodological issues related to $CH_4$ flux upscaling are discussed. The monthly upscaled $CH_4$ flux data product is available for further usage at: https://doi.org/

10.5281/zenodo.2560164.

## 1 Introduction

Methane ($CH_4$) is the second most important anthropogenic greenhouse gas in terms of radiative forcing after carbon dioxide ($CO_2$): 34 times ($GWP_{100}$, including climate-carbon feedbacks) as strong as $CO_2$ according to IPCC (Ciais et al., 2013) and has contributed ~20% of the cumulative GHG related global warming (Etminan et al. 2016). Deriving constraints on its sources

and sinks is thus of utmost importance. The net atmospheric $CH_4$ budget is well constrained by precise $CH_4$ mole fraction measurements around the globe, yet the contribution of individual sources and sinks to this aggregated budget remains poorly understood primarily due to lack of data to constraint the modelling results (Saunois et al., 2016). In order to make accurate predictions of the atmospheric $CH_4$ budget in a changing climate, the response of the various sources and sinks to different drivers needs to be better identified and quantified.

Natural wetlands are the largest and most uncertain source of $CH_4$ to the atmosphere (Saunois et al., 2016). An ensemble of land surface models estimated $CH_4$ emissions from wetlands for the period 2003-2012 to be 185 $Tg(CH_4)$ $yr^{-1}$ (range 153-227 $Tg(CH_4)$ $yr^{-1}$) and for the same period inversion models estimated it to be 167 $Tg$ ($CH_4$) $yr^{-1}$ (range 127-202 $Tg(CH_4)$ $yr^{-1}$)





(Saunois et al., 2016). This discrepancy between bottom-up (process models) and top-down (inversion models) estimates, as well as the range of variability, exemplifies the large uncertainty of the current estimate for natural wetland $CH_4$ emissions. Sources of this uncertainty can be roughly divided into two categories: 1) uncertainty related to the global areal extent of wetlands (e.g. Petrescu et al 2010. Bloom et al., 2017a; Zhang et al., 2016) and 2) uncertainties related to the key $CH_4$ emission

drivers and responses to these drivers (e.g. Bloom et al., 2017a; Saunois et al., 2017). Evaluation of the emission estimates is thus urgently needed, and the results will feed on improvements in process models. Process model improvements will also directly affect the uncertainty of inversion results since they provide important a priori information to the inversion models (Bergamaschi et al., 2013).

Boreal and arctic wetlands comprise up to 50 % of the total global wetland area (e.g. Lehner and Döll, 2004) and these wetlands

make a substantial contribution to total terrestrial wetland $CH_4$ emissions (27 %, based on sum of regional budgets for Boreal North America, Europe and Russia in Saunois et al., 2016). In wetlands, $CH_4$ is produced by methanogenic Archaea under anaerobic conditions and hence the production takes place predominantly under water saturated conditions (e.g. Whalen, 2005). The microbial activity and the resulting $CH_4$ production is thus controlled by quality and quantity of the available substrates, competing electron acceptors and temperature (Le Mer and Roger, 2001). Once produced, the $CH_4$ can be emitted

to the atmosphere via three pathways: ebullition, molecular diffusion through soil matrix, or plant transport. If plants capable of transporting $CH_4$ are present, plant transport is generally the largest of the three (Knoblauch et al., 2015; Kwon et al., 2017; Waddington et al., 1996; Whiting and Chanton, 1992). Importantly, a large fraction of $CH_4$ transported via molecular diffusion is oxidized by methanotrophic bacteria in the aerobic layers of wetland soils and hence never reaches the atmosphere, whereas $CH_4$ transported via ebullition and plant transport can largely bypass oxidation (Le Mer and Roger, 2001; McEwing et al.,

2015). Furthermore, processes related to permafrost dynamics (e.g. thaw, thermokarst processes) and snow cover (snow melt, insulation) have an impact on $CH_4$ flux seasonality and variability in general (Friborg et al., 1997; Helbig et al., 2017; Mastepanov et al., 2008; Zona et al., 2016; Zhao et al 2016). Hence wetland $CH_4$ emissions to the atmosphere are a subtle interplay between water table position, temperature, vegetation composition, methane consumption, availability of substrates and competing electron acceptors.

During the past two decades, eddy covariance (EC) measurements of wetland $CH_4$ emissions have become more common, due to rapid development in sensor technology (e.g. Detto et al., 2011; Peltola et al., 2013, 2014). The latest generation of instruments are rugged enough for long-term field deployment and can function without grid power (McDermitt et al., 2010), increasing the number of sites where $CH_4$ flux measurements can be collected. Due to this progress, EC $CH_4$ flux synthesis studies are now emerging (Petrescu et al., 2015; Knox et al., in review). Similar progress was made with $CO_2$ and energy flux

measurements in the 1990s and now these measurements form the backbone of the global EC measurement network FLUXNET, whose data has provided invaluable insights into terrestrial carbon and water cycles. Some of the most important results have been obtained by upscaling FLUXNET observations using machine learning algorithms to evaluate terrestrial carbon balance components and evapotranspiration (Beer et al., 2010; Bodesheim et al., 2018; Jung et al., 2010, 2011, 2017;





Mahecha et al., 2010). These results are now widely used by the modelling community to evaluate process model performance (e.g. Wu et al., 2017) and to validate satellite-derived carbon cycle data products (e.g. Sun et al., 2017; Zhang et al., 2017a).

In this study, we synthesized EC $CH_4$ flux data from 25 EC $CH_4$ flux sites and developed an observation-based monthly gridded data product of northern wetland $CH_4$ emissions. We focus on northern wetlands (north of 45 °N) due to their significance in

the global $CH_4$ budget and relatively good data coverage and process understanding, at least compared to tropical systems (Knox et al., in review). The Arctic is projected to warm during the next century at a faster rate than any other region, which will likely significantly impact the carbon cycling of wetland ecosystems (Tarnocai, 2009; Zhang et al., 2017b) and permafrost areas of the Arctic-Boreal Region (Schuur et al., 2015). To date, $CH_4$ emission estimates for northern wetlands are typically based on process models (Bohn et al., 2015; Bloom et al., 2017a; Chen et al., 2015; Melton et al., 2013; Stocker et al., 2013;

Wania et al., 2010; Watts et al., 2014; Zhang et al., 2016) or inversion modelling (Bohn et al., 2015; Bruhwiler et al., 2014; Spahni et al., 2011; Thompson et al., 2017; Thonat et al., 2017; Warwick et al., 2016), yet scaling of existing chamber measurements to northern wetland area has also been published (Zhu et al., 2013). However, the first two are not completely independent since the attribution of $CH_4$ emissions derived using inversion models to different emission sources (e.g. wetlands) depends largely on the *a priori* estimates of these emissions (i.e. process models for wetland emissions), highlighting the tight

coupling between these two approaches (Bergamaschi et al., 2013, Spahni et al., 2011). Hence, the main objective of this study is to produce an independent data-driven estimate of northern wetland $CH_4$ emissions. This product could be used as an additional constraint for the wetland emissions and hence aid in process model development. Additionally, the drivers causing $CH_4$ flux variability at the ecosystem scale are also evaluated and methodological issues are discussed which will support future $CH_4$ wetland flux upscaling studies.

**2 Materials and Methods**

Data from flux measurement sites (Fig. 1) were acquired and used together with forcing data to estimate $CH_4$ emissions from northern wetlands with monthly time resolution using a random forest (RF) approach. Both in-situ measurements and remote sensing are utilized in this study. In this section, the RF approach is briefly introduced (Sect. 2.1) and data selection, quality filtering, gap filling and aggregation to monthly values are described (Sect. 2.3). After this procedure, 40.7 site-years were

available for the analysis. To perform upscaling to all wetlands north of 45 °N, gridded data products of the flux drivers are needed, as well as wetland distribution maps. These products are presented in Sect. 2.4 and 2.5, respectively. Finally, the upscaled wetland $CH_4$ emissions are compared against process model outputs, with the models briefly described in Sect. 2.6. Here, wetlands are defined as terrestrial ecosystems with water table positions near the land surface and with plants that have adapted to these water-logged conditions. We exclude lakes, reservoirs and rivers from the study, in addition to ecosystems

with significant human influence (e.g. drainage, rewetting). We consider peat forming wetlands (i.e. mires), which can be further classified as fens and bogs based on hydrology, as well as wetlands with hydric mineral soils. Tundra wetlands may have only a shallow peat layer, or none at all. Unified classifications for wetlands are still lacking, and typically different



countries follow their own classification scheme, albeit some joint classification schema have been developed (e.g. Ramsar Classification System for Wetland Type).

## 2.1 Random forest algorithm

Random forest (RF) is a machine-learning algorithm that can be used for classification or regression analyses (Breiman, 2001) and in this study the RF models consists of a large ensemble of regression trees. Each individual regression tree is built by training it with a random subset of training data and the trees are trained independently of each other. The RF model output is then the average of all the predictions made by individual regression trees in the forest. Hence the RF algorithm applies the bootstrap aggregation (bagging) algorithm and takes full advantage of the fact that ensemble averaging decreases the noise of the prediction. In addition to random selection of training data, the predictor variables used in split nodes are also selected from a random sample of all predictors which minimizes the possible correlation between trees in the forest (Breiman, 2001) and decreases the possibility of overfitting.

Performance of RF algorithms to predict $CO_2$ and energy fluxes across FLUXNET sites were compared against other machine-learning algorithms such as artificial neural networks and multivariate regression splines by Tramontana et al. (2016) who showed that differences between methods were negligible. These results are also likely to apply for $CH_4$ fluxes. For a thorough description of the RF algorithm for flux upscaling purposes, the reader is referred to Bodesheim et al. (2018) (and references therein).

In this study, the RF models were developed using the MATLAB 9.4.0 (R2018a) TreeBagger function with default values similarly to Bodesheim et al. (2018). These settings included a minimum of five samples in a leaf node and used MSE as a metric for deciding the split criterion in split nodes. Each trained forest consisted of 300 randomized regressions trees.

### 2.1.1 RF model development for $CH_4$ flux gapfilling

Our RF algorithm was used for gapfilling the $CH_4$ flux time series at a daily time step, and the performance of the RF model was evaluated against so-called out-of-bag (OOB) data (approximately 1/3 of data for each tree). Since each individual tree in the RF model was trained using a subset of training data, the rest of the data (i.e. OOB data) can be used as independent validation data to evaluate the prediction performance of that particular regression tree and hence the whole forest (Breiman, 2001). Only the five most important predictors were retained for the gapfilling models for each site. The relative importance of predictors (e.g. air temperature and the others) was evaluated by randomly shuffling the predictor data and then estimating the increase in mean squared error (MSE) when model output is compared against OOB data (Breiman, 2001). For important predictors MSE will increase significantly due to shuffling, whereas the effect of shuffling the less important predictors on MSE is minor. Note that this procedure was executed separately for each site and thus different predictors may have been used for different sites for gapfilling.



### 2.1.2 RF model development for CH₄ flux upscaling

For upscaling purposes, one RF model was developed using all the available data in order to maximize the information content derived from the available data for the upscaled CH₄ flux map. The model performance or uncertainty, however, was evaluated by using the data in two ways. The predictive performance of the model was assessed using the widely used leave-one-site-out cross-validation scheme (e.g. Jung et al., 2011). In order to avoid correlation between training data and validation data, sites were excluded from the training data when a site located nearby (closer than 100 km) was used as a validation site (Roberts et al., 2016). In turn, the uncertainty of the upscaled fluxes was estimated by bootstrapping. 200 independent RF models were trained using a bootstrap sample of available data. This yielded 200 predictions for each pixel and time step in the upscaled CH₄ flux map and the variability over this prediction ensemble was used as an uncertainty measure. This follows the methodology used e.g. by Aalto et al. (2018) and Zhu et al. (2013). One should note that this uncertainty estimate reflects the ability of the RF model to capture the dependence of CH₄ flux on the used predictors in the available data, however it does not have any reference to actual in-situ CH₄ fluxes unlike the model predictive performance estimated with cross-validation.

Predictors for the RF model used in upscaling were determined following Moffat et al. (2010). First, the RF models were trained for each site using one predictor at a time (see all the predictors in Table 1). The single predictor which yielded the best match against validation data (leave-one-site-out scheme) was selected as the primary driver. Then, the RF models were trained again with the primary driver plus each of the other predictors in turn as secondary drivers. Then the RF model performance was again evaluated, and the best predictor pair selected for the next round. This procedure was continued until all the predictors were included in the RF model. The smallest set of predictors capable of producing adequate RF model performance was used for flux upscaling.

### 2.2 Metrics for model performance evaluation

The RF model performance was evaluated against independent validation data using a set of statistical metrics, which were related to different aspects of model performance. During the RF model training the mean squared error (MSE) was optimized:

$$MSE = \overline{(o - p)^2}, \tag{1}$$

where $o$ and $p$ are vectors containing the observed and predicted values, respectively, and the overbar denotes averaging.

The Nash-Sutcliffe model efficiency (Nash and Sutcliffe, 1970) (NSE) can be used to evaluate how well the model is able to predict validation data when compared against a reference (typically the mean of the validation data):

$$NSE = 1 - \frac{\sum_{i=1}^{n}(o_i - p_i)^2}{\sum_{i=1}^{n}(o_i - \bar{o})^2}, \tag{2}$$

where $i$ is index running over all the $n$ values in the $o$ and $p$ vectors. When NSE is equal to 1, there is a perfect match between prediction and observations. Values above 0 imply that the model predicts the observations better than the mean of observations and values below 0 indicate that the predictive capacity of the model is worse than the mean of validation data. Note that NSE calculated with Eq. (2) above is equivalent to the coefficient of determination calculated using residual sum of squares and total sum of squares. However, following the approach used in previous upscaling studies (e.g. Bodesheim et al., 2018;



Tramontana et al., 2016), we opted to call this metric NSE. Instead, the coefficient of determination ($R^2$) was estimated as the squared Pearson correlation coefficient. Note that $R^2$ and NSE are equal when there is no bias between $o$ and $p$ and the residuals follow Gaussian distribution. In the Results section Pearson correlation coefficients obtained with different model runs are compared using Fisher's r to z transformation.

The standard deviation ($\sigma$) of the model residuals was used to evaluate the spread of model residual values (RE):

$$RE = \sigma(o - p), \tag{3}$$

whereas bias between model predictions and validation data were used to estimate the systematic uncertainty in the upscaled fluxes (BE):

$$BE = \overline{o - p}. \tag{4}$$

Note that RE equals RMSE when there is no systematic difference between the model predictions and observations (i.e. when BE equals zero).

## 2.3 Data

### 2.3.1 Data from eddy covariance flux measurement sites

Data were acquired from 25 sites that 1) measure $CH_4$ fluxes with the EC technique, 2) are located north of 45 °N and 3) are
wetlands without substantial human influence on ecosystem functioning (see the site locations in Fig. 1 and the site list in Appendix A). The sites were evenly distributed among fens (9 sites), bogs (7) and wet tundra (9) ecosystems across tundra (11), boreal (8) and temperate (6) biomes, as defined in Olson et al., (2001). At 15 of the 25 sites, sedges (e.g., *Rhynchospora alba, Eriophorum vaginatum, Carex limosa*) were the dominant vascular plant functional type in the flux measurement source area. Most of the sites (18 out of 25) were located north of 60 °N and the highest density of sites were in Fennoscandia and
Alaska (Fig. 1). The magnitude of monthly $CH_4$ flux data varied between sites and the median time series length was 14.5 months of $CH_4$ flux data per site. The sites represent northern wetlands sufficiently well to create a first upscaled $CH_4$ flux product based on EC data. In the Results section, sites are referred to with their FLUXNET IDs and if not available, new temporary site IDs were generated for the usage in this study (see Appendix A).

Site PIs provided $CH_4$ fluxes and their potential drivers (air temperature and pressure, precipitation, wind speed and direction,
friction velocity, net ecosystem exchange of $CO_2$ and its components, photosynthetically active radiation, water table depth, soil temperature) and a description of each site. However, out of the in-situ measurements only air temperature and precipitation were used for developing the RF model for flux upscaling since gridded data products of the other drivers were not readily available and/or the data for the other drivers were missing from several sites.

Thirty-minute-averaged flux data were acquired from 21 sites and daily data were provided for four sites. The flux time series
were quality filtered by removing fluxes with the worst quality flag (based on 0,1,2 flagging scheme, Mauder et al., 2013) and friction velocity filtered using site-specific threshold if it was typically done at the flux site and friction velocity data were available. After filtering, daily medians were calculated if the daily data coverage was above 29 out of 48 half-hourly data



points (daily data coverage at minimum 10 data points for sites without diel pattern in CH$_4$ flux) and no gap-filling was done to the time series prior to calculation of daily values. While this may cause slight systematic bias in the daily flux values, this bias is unlikely to be significant because the magnitude of diel patterns in CH$_4$ fluxes is typically moderate (e.g. Long et al., 2010) or negligible (e.g. Rinne et al., 2018), although at sites with Phragmites cover a relatively strong diurnal cycle can be
observed (e.g. Kim et al., 1999; Kowalska et al., 2013).

Unlike the CH$_4$ flux data, the other in-situ data from the sites were gap-filled prior to the calculation of daily values. The gapfilling was done only if the daily data coverage was above 60 % and for the days with lower data coverage daily values were not calculated. Shorter gaps (<2 hours) were filled with linear interpolation, whereas longer gaps (between 2 to 14.5 hours) were replaced with mean diurnal variation within a 30-day moving window. However, for precipitation daily sums were
calculated without any gapfilling. Besides the measurements at the sites, potential solar radiation (R$_{pot}$) and its time derivative (der(R$_{pot}$)) were calculated based on latitude and time of measurement. In order to remove the R$_{pot}$ latitudinal dependence it was normalized to be between 0 and 1 before usage.

CH$_4$ flux drivers measured in-situ, in addition to the remote sensing data (Sect. 2.3.2), were used for the gapfilling of CH$_4$ time series with the RF algorithm (Sect. 2.1.1). For each site the gapfilling models generally agreed well with the independent
validation data (mean NSE=0.74 and mean RMSE = 9 nmol m$^{-2}$ s$^{-1}$). After gapfilling, the CH$_4$ flux time series were aggregated to monthly values if the monthly data coverage prior to gapfilling was at least 20 %.

The daily time series of air temperature and precipitation measured at the sites were gapfilled using the WFDEI (WATCH Forcing Data methodology applied to ERA-Interim data) data (Weedon et al., 2014). Prior to using the WFDEI data for gapfilling, the data were bias corrected for each site as is typically done for climate or weather reanalysis data (e.g. Räisänen
& Räty, 2013; Räty et al., 2014). For precipitation, the mean of WFDEI data were simply adjusted to match site mean precipitation. For air temperature the bias correction was done for each month separately using quantile mapping with smoothing within a moving seven-month window. Quantile mapping compares the cumulative distribution functions (CDFs) of WFDEI and site measurements against each other and adjusts the WFDEI data so that after adjustment its CDF matches with the CDF of the site measurements. See more details about the bias correction procedures e.g. in Räisänen & Räty (2013).
After gapfilling the daily time series with WFDEI data, monthly and annual precipitation were calculated, in addition to monthly mean air temperature.

### 2.3.2 Remote sensing data

Data from the Moderate Resolution Imaging Spectrometer (MODIS) were used in this study. For RF model development the following data products at 500 m or 1000 m spatial resolution were used: MOD10A1 snow cover (Hall and Rigs, 2016),
MOD11A2 daytime and night-time land surface temperature (LSTd and LSTn, Wan et al., 2015), MOD13A3 enhanced vegetation index (EVI, Didan, 2015) and MOD09A1 surface reflectance (Vermote, 2015). More elaborate satellite products estimating ecosystem GPP and net primary productivity (NPP; MOD17) were not included here for two reasons: 1) many of the sites included here were misclassified in the land cover map used in MOD17 e.g. as woody savanna, hence severely



influencing the estimated GPP and NPP (Zhao et al., 2005), and 2) sites that were correctly classified as permanent wetlands were in fact assigned a fill value and removed from the product since the product is not strictly valid for these areas (Lees et al., 2018). All the remote sensing data products were quality filtered using the quality flags provided along with the data.

The MODIS snow cover ranged from 0 (no snow) to 100 (full snow cover) and was converted to a simple snow cover flag consisting of 0 and 1 depending whether the snow cover data were below or above 50, respectively. A vector containing days since snow melt (DSSM) was calculated using the snow cover flag and normalized to 0 (beginning) and 1 (end) for each growing season (Mastepanov et al., 2013). The MOD09A1 surface reflectance at bands 2 (841-876 nm) and 5 (1230-1250 nm) were used to calculate the simple ratio water index (SRWI=band 2/band 5) following Zarco-Tejada & Ustin (2001). SRWI showed spurious values when there was snow cover and hence these points were replaced with the mean SRWI observed at each site when there was no snow. Meingast et al. (2014) showed that SRWI can be used as a proxy for wetland water table depth, although their results were affected by changes in vegetation cover, which might hinder across-site comparability in this study. Additionally, following the temperature and greenness modelling approach (Sims et al., 2008), a product of EVI and LSTd was included in the analysis as a proxy for gross primary productivity (GPP), following a previous peatland study (Schubert et al., 2010). The remote sensing data were provided with daily (MOD10A1), 8-day (MOD09A1, MOD11A2) or monthly (MOD13A3) time resolution and the data were aggregated to monthly means prior to usage.

### 2.3.3 Additional categorical variables

The sites were also classified based on the presence of permafrost in the source area (present or absent) and according to biome type. Furthermore, the data were categorized based on wetland type and sedge cover as in Treat et al. (2018) and Turetsky et al. (2014). However, such information is not available in the gridded format needed for upscaling, nevertheless inclusion of these variables can be used to assess how much they increase the predictive performance of the model. Biome types (temperate, boreal, tundra) were determined from Olson et al. (2001).

### 2.4 Gridded data sets used in flux upscaling

For upscaling $CH_4$ fluxes using the developed RF model, the LST data were acquired from the aggregated product MOD11C3 (Wan et al., 2015), and snow cover data from MOD10CM (Hall and Riggs, 2018). Distribution of permafrost in the northern latitudes were estimated using the circum-Arctic map of permafrost derived by National Snow and Ice Data Center (Brown et al., 2002). The resolution of the gridded data was adjusted to match the resolution of the wetland maps using bilinear interpolation if needed. Additionally, land and ocean masks (Jet Propulsion Laboratory, 2013) were utilized when processing the gridded data sets.

### 2.5 Wetland maps

Upscaled fluxes were initially estimated in flux densities per wetland area, that is (amount of $CH_4$) per (area of wetland) per (unit of time). To create a gridded product of $CH_4$ emissions from the northern wetlands, these upscaled flux densities were



converted into (amount of $CH_4$) per (grid cell area) per (unit of time) using wetland maps. Wetland mapping is an ongoing field of research and the usage of different wetland maps contributes to the uncertainty of global wetland $CH_4$ emission estimates (e.g. Bloom et al., 2017a; Zhang et al., 2017b). Hence, three different wetland maps (PEATMAP, DYPTOP and GLWD) were used in this study to evaluate how much they affect the overall estimates of northern wetland $CH_4$ emissions.

The recently developed static wetland map PEATMAP (Xu et al., 2018) combines detailed geospatial information from various sources to produce a global map of wetland extent. Within the presented study, the polygons in PEATMAP were converted to fractions of wetland in 0.5° grid cells. While PEATMAP is focused on mapping peatlands, marshes and swamps (typically on mineral soil) are included in the product for certain areas in the northern latitudes. However, most of the wetlands in the northern latitudes are peatlands and thus PEATMAP is suitable for our upscaling purposes. The dynamic wetland map

estimated by the DYPTOP model (Stocker et al., 2014) was used by aggregating peat and inundated areas to form one dynamic wetland map with 1° resolution. The widely used Global Lakes and Wetlands Database (GLWD, Lehner and Döll, 2004) is a static wetland map with 30 arc second resolution and was used in this study as a point of reference for the other two maps. The map was aggregated to 0.5° resolution and lakes, reservoirs and rivers were excluded from the aggregated map.

## 2.6 Process models

The upscaled $CH_4$ fluxes were compared against two process models: LPX-Bern (Spahni et al., 2013; Stocker et al., 2013; Zürcher et al., 2013) and the model ensemble WetCHARTs version 1.0 (Bloom et al., 2017a, 2017b). LPX-Bern is a dynamic global vegetation model which models carbon and nitrogen cycling in terrestrial ecosystems. The model has a separate peatland module with peatland-specific plant functional types (see more details in Spahni et al., 2013). The wetland extent in LPX-Bern was dynamically estimated using the DYPTOP approach with 1° resolution (Stocker et al., 2014). WetCHARTs combines

several prescribed wetland maps with different gridded products for heterotrophic respiration and temperature sensitivity ($Q_{10}$)-parameterizations for $CH_4$ production to form a model ensemble of wetland $CH_4$ emissions (Bloom et al., 2017b). Here we used the extended ensemble of WetCHARTs.

## 3 Results

### 3.1 Selecting the predictors for the RF model

The predictors in Table 1 were selected one-by-one using the procedure described in Sect. 2.1.2. The order in which the predictors were selected is shown in Fig. 2. LSTn alone gave NSE=0.29. After including the category presence or absence of permafrost, $R_{pot}$, SC and biome class increased NSE to 0.47, albeit the influence of SC and biome class on the model performance was marginal. Additional predictors did not increase the model performance further because 1) they were strongly correlated with a predictor already included in the model (e.g. $T_{air}$ is correlated with LSTn) and hence they did not add any

new information to the system, or 2) the predictors did not contain any information about $CH_4$ flux variability. The model response to other predictors than biome category was physically reasonable (e.g. permafrost and snow cover decrease fluxes,





close to exponential dependence on LSTn), whereas the response to biome category was contrary to expectations. The RF model estimated the CH$_4$ flux magnitude from the different biomes to be in the order tundra<temperate<boreal, whereas in prior studies it has been shown to be in the order tundra<boreal<temperate (Knox et al., in review Treat et al., 2018; Turetsky et al., 2014). This discrepancy may be due to the limited number of measurement sites and related sampling bias problems.

Hence in order not to upscale an incorrect pattern of decreasing CH$_4$ emissions when moving from boreal to temperate regions, the biome class was omitted from upscaling. In the subsequent analysis and flux upscaling only the four first predictors (LSTn, permafrost category, R$_{pot}$ and SC) are utilized.

We further tested whether information about wetland type or sedge cover would improve the model performance, although these variables were not available in gridded format and hence were not usable for upscaling. Including the sedge flag increased

the NSE to 0.53, although the increase in Pearson correlation was not statistically significant (p>0.05, comparison of correlation coefficients using Fisher's r to z transformation). Also, wetland type did not have a statistically significant influence on the model performance (p>0.05 and NSE=0.49 if type included). Using many categorical variables in a RF model may be problematic because each site may end up with a unique combination of categorical variables.

The most important predictor for the model was temperature, similar to numerous studies showing that wetland CH$_4$ emissions

are strongly correlated to soil temperature (Christensen et al., 2003; Helbig et al., 2017; Jackowicz-Korczyński et al., 2010; Rinne et al., 2018; Yvon-Durocher et al., 2014). Estimating apparent Q$_{10}$ from the RF model LSTn dependence yielded a value of 1.90+/-0.03 and for validation data it was slightly higher (1.97+/-0.06) (Fig. 3). These values are comparable to the ones reported in Turetsky et al. (2014) for CH$_4$ chamber measurements at bog and fen sites. The temperature dependence of CH$_4$ production is modelled in many process models with the parameter Q$_{10}$ value close to 2 (Xu et al., 2016b), which agrees with

the CH$_4$ emission temperature dependence shown here. However, one should note that also CH$_4$ oxidation depends on temperature and the derived apparent Q$_{10}$ value describes the temperature dependence of surface CH$_4$ emission, which is always a combination of CH$_4$ production and oxidation.

## 3.2 Model agreement with validation data

The overall systematic bias (BE) between the RF predictions and validation data was negligible (Fig. 4), whereas the spread

of the data (RE) was more pronounced. RE is evident in Fig. 4 also as significant scatter around the 1:1 line. Following Moffat et al. (2010), RE was analysed further by binning the data based on CH$_4$ flux magnitude and calculating RE for each bin. RE clearly correlated with flux magnitude (RE = (0.52±0.06)FCH$_4$+(3.3±2.0) nmol m$^{-2}$ s$^{-1}$, where FCH$_4$ denotes CH$_4$ flux) indicating that the relative random error of the RF model prediction was nearly constant and approximately 50 % for high fluxes. The systematic error BE did not show a clear dependence on flux magnitude. The RF model performance was worse

on site mean level than with monthly data. When comparing site means, NSE and R$^2$ were both 0.25 and RE and BE were 27.0 nmol m$^{-2}$ s$^{-1}$ and 1.5 nmol m$^{-2}$ s$^{-1}$, respectively. Possible drivers causing the remaining CH$_4$ flux variability not captured by the RF model (i.e. the scatter in Fig. 4) are discussed in Sect. 4.2.1.



When considering the model performance for each site separately, the agreement shows different characteristics (see Fig. 5 for four examples). For individual sites the magnitude of BE is typically somewhat higher (median of absolute value of BE approximately 11 nmol m$^{-2}$ s$^{-1}$), whereas RE is lower than for the overall agreement (median RE approximately 10 nmol m$^{-2}$ s$^{-1}$). These results indicate that the upscaled CH$_4$ fluxes have in general relatively low bias and high random error, whereas

individual pixels in the upscaled CH$_4$ map may have higher bias, but lower random error.

The mean annual cycle of CH$_4$ emission predicted by the RF model agrees well with the mean annual cycle calculated from the validation data (not shown). During the nongrowing season the RF model slightly overestimates the fluxes (15 % overestimation), but during rest of the year the differences are negligible (<1 %). However, for individual sites the agreement is not as good. For instance, at US-Los (located in Wisconsin, US) the modelled CH$_4$ emissions start to increase one month

too early in the spring (Fig. 5b) and the nongrowing season fluxes are overestimated at all four example sites (FI-Sii, US-Los, US-Atq and RU-Ch2; Fig. 5). Out of the example sites, the mean flux magnitude is modelled well at FI-Sii (Fig. 5a), whereas at US-Los (Fig. 5b) and US-Atq (Fig. 5c) the RF model overestimates and at RU-Ch2 (Fig. 5d) underestimates the CH$_4$ emissions. The flux bias had a relatively large impact on site-specific NSE. For example, for US-Atq NSE was -1.85, meaning that the mean of observations would be a better predictor for this site than the RF model (see the NSE definition in Sect. 2.2).

The RF model is not able to replicate the interannual variability in CH$_4$ emissions at the example sites and explaining the interannual variability has been difficult also in previous upscaling studies of CO$_2$ and energy fluxes (e.g. Tramontana et al., 2016).

In general, the RF model performance was better for sites without than with permafrost (r = 0.66 and r = 0.51, respectively; p<0.05), which is likely related to the fact that at sites with permafrost the MODIS LSTn is not as directly related to the soil

temperature than at sites without permafrost. Hence LSTn is not as good proxy for the temperature which is controlling both CH$_4$ production and consumption and this results in a worse performance than at sites without permafrost.

### 3.3 Upscaled CH$_4$ fluxes

The RF model was used together with the gridded input datasets (Sect. 2.4) and wetland maps (Sect. 2.5) to estimate CH$_4$ emissions from northern wetlands during years 2013 and 2014. The mean CH$_4$ emissions of the two years from the RF model are plotted in Fig. 6 together with CH$_4$ wetland emission maps from process model LPX-Bern and model ensemble

WetCHARTs. Differences between the process model estimations and upscaled fluxes are shown in Fig. 7. In general, the spatial patterns look similar in all emission maps, which is understandable since the spatial variability is largely controlled by the underlying wetland distributions. One noteworthy difference is that WetCHARTs, RF-PEATMAP (i.e. RF modelling with PEATMAP) and RF-GLWD show higher emissions from western Canada than LPX-Bern or the upscaled fluxes using the wetland map from that process model (RF-DYPTOP). The other difference is RF-GLWD show negligible emissions from

Fennoscandia (Fig. 6c). These differences are related to differences in the underlying wetland maps.

The uncertainties of the upscaled fluxes were estimated from the spread of predictions made with the ensemble of 200 RF models (Sect. 2.1.2) and are shown in Fig. 8. The uncertainty mostly scales with the flux magnitude (compare Fig. 6 a)-c) with



Fig. 8 a)-c)), meaning that grid cells with high fluxes tend to have also high uncertainties. However, the relative flux uncertainty does have some geographical variation (Fig. 8 d)-f)). The highest relative uncertainties are typically at the highest and lowest latitudes of the study domain. In these locations the dependencies between the predictors and the $CH_4$ flux are not as well-defined as in the locations with lower uncertainties leading to larger spread in the ensemble of RF model prediction. For instance, at low latitudes LSTn may go beyond the range of LSTn values in the training data (see the range in Fig. 3) and hence the RF model predictions are not well-constrained in these situations. On the other hand, lower relative uncertainties are typically obtained for locations close to the measurement sites incorporated in this study (compare Fig. 1 and 8), since the dependencies between the predictors and the $CH_4$ flux are defined better.

The seasonality of the upscaled fluxes and $CH_4$ fluxes from process models are similar with highest $CH_4$ emissions in July-August and lowest in February, and this seasonal pattern is comparable throughout the whole study domain (Fig. 9). Warwick et al. (2016) and Thonat et al. (2017) showed that the northern wetland $CH_4$ emissions should peak in August-September in order to explain correctly the seasonality of atmospheric $CH_4$ mixing ratios and isotopes measured across the Arctic. Hence the wetland $CH_4$ emissions presented here are peaking approximately one month too early to perfectly match with their findings. $CH_4$ flux magnitude agrees well between WetCHARTs and the upscaled flux during spring and midsummer (April-July), whereas LPX-Bern is estimating lower fluxes (0 % and 26 % difference, respectively). During late summer and autumn (August-October) both process models are estimating slightly lower fluxes than the upscaled estimate (17 % and 19 % difference, respectively). The upscaled fluxes show somewhat higher emissions also during the nongrowing season (November-March) than the two process models (27 % and 35 % difference, see Table 2) and the upscaled estimates of nongrowing season emissions are relatively close to the recent model estimate by Treat et al. (2018). This result promotes the recent notion that process models might be underestimating nongrowing season fluxes at high latitudes (e.g. Treat et al., 2018; Xu et al., 2016a; Zona et al., 2016).

Treat et al. (2018) adjusted WetCHARTs model output so that it matches with their estimates of nongrowing season $CH_4$ emissions and then estimated annual wetland $CH_4$ emissions north from 40 °N to be $37 \pm 7$ Tg($CH_4$) yr$^{-1}$ using this adjusted model output. The estimates derived here for the annual emissions using the three wetland maps are similar (see Table 2), especially when considering that here we have slightly smaller study domain (above 45 °N). The two process models included in this study estimated slightly lower mean annual emissions than the upscaled fluxes (11 % and 26 % difference between the mean upscaled estimate and WetCHARTs and LPX-Bern, respectively; see also Table 2). However, given the uncertainties in upscaling as well as in process models this can be regarded as relatively good agreement.

In order to further evaluate the agreement between the upscaled fluxes and process models we concentrated on two specific regions: Hudson Bay lowlands (HBL) and western Siberian lowlands (WSL) (see locations in Fig. 1). The upscaled fluxes show clearly higher annual emissions for both subdomains than the two process models, or what has been previously estimated in the literature (Table 2), although for WSL the upscaled estimates are within the range of variability observed between process models and inversion modelling in WETCHIMP-WSL (Bohn et al., 2015) and close to Thompson et al. (2017). Bohn et al. (2015) also notes that the upscaled estimate by Glagolev et al. (2011) is most likely an underestimate of the $CH_4$ emissions



from the WSL area. Furthermore, the process models in Bohn et al. (2015) are likely underestimating the nongrowing season $CH_4$ emissions which might partly explain the discrepancy to the upscaled estimates in this study. Hence, the upscaled $CH_4$ emission estimates for the WSL area, while large, are still in a reasonable range.

For HBL, the discrepancy between upscaled emission estimates and the estimates based on process models or previous studies

is larger (Table 2). The upscaling results agree with Zhang et al. (2016) and Melton et al. (2013) but show over twice larger emissions from HBL than the other estimates (Table 2). This cannot be explained by wetland mapping since the difference holds also when DYPTOP wetland map is used in upscaling. There are not many long-term EC flux studies conducted in the HBL area and the only one found (Hanis et al., 2013) showed on average 6.9 g($CH_4$) m$^{-2}$ annual emissions at a subarctic fen located in the HBL. If the upscaled $CH_4$ emissions are downscaled back to ecosystem level at the HBL area with wetland maps,

we get on average 11.0 g($CH_4$) m$^{-2}$ annual $CH_4$ emission for the HBL area based on the RF model output, which is 1.6 times larger than the estimate by Hanis et al. (2013). While Hanis et al. (2013) studied only one wetland during different years than here (years 2008…2011 in Hanis et al. (2013), here 2013…2014) it is still noteworthy that the relative difference between Hanis et al. (2013) and this study is similar to the discrepancy between this study and the inversion estimates (Pickett-Heaps et al. (2011); Thompson et al. (2017)) at the whole HBL scale. All three studies (Hanis et al. (2013); Pickett-Heaps et al.

(2011); Thompson et al. (2017)) show near zero $CH_4$ emissions during October…April and onset of $CH_4$ emissions in mid-May or even June, largely dependent on when the ground was free of snow and unfrozen. This is somewhat surprising given the fact that only 32 % of wetlands in the area are underlain by permafrost (based on amalgam of PEATMAP and permafrost map) and hence the soils are likely not completely frozen and some nongrowing season $CH_4$ emissions are likely to occur in such conditions (e.g. Treat et al., 2018). The upscaled $CH_4$ emissions show on average 1.1 Tg($CH_4$) yr$^{-1}$ emissions during these

nongrowing season months for the HBL area. This partly, but not completely, explains the discrepancy between the $CH_4$ emission estimates for the HBL area. All these results suggest that the upscaled product likely overestimates $CH_4$ emissions from the HBL area.

## Discussion

### 4.1 Comparing the RF model predictive performance to previous studies

The RF model performance was worse when compared against independent validation data than what has been achieved in previous upscaling studies for GPP and energy fluxes (R$^2$>0.7), and ecosystem respiration (Reco; R$^2$>0.6), whereas the performance for net ecosystem exchange of $CO_2$ (NEE) has been similar (R$^2$<0.5) as here for monthly $CH_4$ emissions (e.g. Jung et al., 2010; Tramontana et al., 2016). Likely reasons for this finding include for instance that for other fluxes there is simply more data available from several sites spanning the globe, whereas here we have data from 25 sites with $CH_4$ fluxes.

Furthermore, the drivers (or proxies for the drivers) of e.g. GPP and energy fluxes are more easily available from remote sensing (e.g. MODIS) and weather forecasting re-analysis data sets (e.g. WFDEI), whereas $CH_4$ emissions are more related to processes taking place belowground and hence the drivers for these processes are more difficult to measure remotely, which



is in practice needed for the upscaling. Also, there are temporal lags between changes in drivers (e.g. LSTn) and $CH_4$ flux responses to these changes and hence training a machine learning model such as RF model on such data is difficult since RF model assumes direct relationship between the change and response. However, one should also note that GPP or Reco are never directly measured with the EC technique, they are always at least partly derived products (Lasslop et al., 2009; Reichstein

et al., 2005). Hence direct functional relationships between GPP and Reco and environmental drivers are inherently included in these flux estimates, whereas NEE and $CH_4$ emissions can be directly measured without additional modelling. Also, both NEE and $CH_4$ emissions are combinations of two counteracting processes (NEE: GPP and Reco; $CH_4$ flux: production and oxidation). Therefore, GPP and Reco upscaling algorithms show better correspondence with validation data than for NEE or $CH_4$ emissions and the results for NEE would be the correct point of reference for the RF model performance presented here.

While the RF model performance in this study was inferior to previous upscaling studies for other fluxes, it was still comparable to what has been shown before for several process models for $CH_4$ emission (McNorton et al., 2016; Wania et al., 2010; Zürcher et al., 2013; Zhu et al., 2014; Xu et al., 2016a). For instance, McNorton et al. (2016) validated the land-surface model JULES against $CH_4$ flux data from 13 sites and found $R^2=0.10$ between the validation data and the model. Wania et al. (2010) found on average RMSE=29 nmol m$^{-2}$ s$^{-1}$ and RMSE=42 nmol m$^{-2}$ s$^{-1}$ with and without tuning their model LPJ-WhyMe against

$CH_4$ flux data from seven sites. Zürcher et al. (2013) found the time-integrated $CH_4$ flux to be well represented by LPX-Bern model across different sites and a tight correlation ($R^2 = 0.92$) is found between simulated and measured cumulative site emissions after calibrating the model against the measurements. While Xu et al. (2016a) did not explicitly show any statistical metrics, their model (CLM4.5) comparison against site level $CH_4$ flux data seemed to be somewhat better than in Wania et al. (2010) or McNorton et al. (2016). Xu et al. (2016a) emphasize the importance of nongrowing season emissions and the fact

that their model was clearly underestimating these emissions. Zhu et al. (2014a) calibrated their model TRIPLEX-GHG for each measurement site by changing e.g. the $Q_{10}$ for $CH_4$ production and $CH_4$ to $CO_2$ release ratio to be site-specific and found on average $R^2=0.64$ when comparing the calibrated model against measurements at 17 $CH_4$ flux measurement sites. However, their findings are not directly comparable to the RF model agreement with validation data shown here due to their model calibration against data before comparison. Nevertheless, their results show that even after calibration, the process models are

not fully able to capture the $CH_4$ flux variability in measurements. Miller et al. (2014) argued that some of the process models might be too complex so that their required input information cannot be reliably provided at larger scales. All of these five models (JULES, LPJ-WhyMe, LPX-Bern, CLM4.5 and TRIPLEX-GHG) are contributing to the global $CH_4$ budget estimation within the Global Methane Project (Saunois et al., 2016), highlighting that these results summarize the agreement between state-of-the-art process models and field measurements.



## 4.2 Methods to improve RF model predictive performance

### 4.2.1 Missing predictors

In this study a statistical model was developed using the RF algorithm, and the model was able to yield $R^2$=0.47 against monthly $CH_4$ flux validation data. The incomplete match between the RF model and validation data is likely caused by the fact that not all the possible drivers causing within and across site variability to the $CH_4$ emissions were included in the analysis and hence all the variability could not be explained by the model.

Christensen et al. (2003) were able to explain practically all the variability in annual $CH_4$ emissions in their multisite study with only two predictors: temperature and the availability of substrates for $CH_4$ production. Also, Yvon- Durocher et al. (2014) speculate that the amount of substrates for microbial $CH_4$ production explains across site variability of $CH_4$ fluxes in their data. However, such data about substrates is impossible to achieve in a gridded format, which is a strict requirement for upscaling. Hence proxies for the substrates available for methanogenesis are needed. The current paradigm on wetland $CH_4$ emissions is that most of the emitted $CH_4$ is produced from recently fixed carbon, since $CH_4$ producing Archaea favour fresh labile carbon (e.g. Chanton et al., 1995; Whiting and Chanton, 1993). Most of the process models are based on the premise that a certain fraction of ecosystem net primary productivity (NPP) is available and used for $CH_4$ production or alternatively a fraction of heterotrophic respiration is allocated to $CH_4$ emissions (e.g. Xu et al., 2016b). Hence, hypothetically ecosystem NPP (or GPP) could also be included as a predictor here for the RF model and used as a proxy for the amount of substrates available for $CH_4$ production. However, the RF model performance in this study was not enhanced if variables closely related to NPP (EVI and the product of EVI and LSTd) were included as predictors. Also, Knox et al. (in review) did not find GPP as an important predictor of $CH_4$ emission variability in their multi-site synthesis study.

Using the ecosystem level NPP (or proxies for it) for the RF model development might be an overly simplified approach, since it has been shown that it is especially the deep-rooting sedges and their NPP that are important for $CH_4$ production (Joabsson and Christensen, 2002; Ström et al., 2003, 2012; Waddington et al., 1996). Hence, information about plant functional types (PFTs) would be needed to better explain the $CH_4$ flux variability (Davidson et al., 2017; Gray et al., 2013). Furthermore, the fraction of the fixed carbon allocated to the roots and released as root exudates (hence available for $CH_4$ production) varies between species and root age (Proctor and He, 2017; Ström et al., 2003), further complicating this issue. The sedges also act as conduits for $CH_4$ allowing the $CH_4$ produced below water level to rapidly escape to the atmosphere and bypass the oxic zone in which the $CH_4$ might have otherwise been oxidized (Waddington et al., 1996; Whiting and Chanton, 1992). Besides sedges, *Spaghnum* mosses are also important because methanotrophic bacteria that live in symbiosis with these mosses significantly decrease the $CH_4$ emissions to the atmosphere when they are present (Larmola et al., 2010; Liebner et al., 2011; Parmentier et al., 2011; Raghoebarsing et al., 2005). In a modelling study, Li et al. (2016) showed that it was essential to consider the vegetation differences between sites when modelling $CH_4$ emissions from two northern peatlands. Hence, ideally one should have wetland species composition in a gridded format together with their NPP across the high latitudes to significantly improve the upscaling results from the results shown here. Naturally such information is not readily available and



therefore alternatively modelled estimates could be used (e.g. LPX-Bern which includes several peatland-specific PFTs which are allowed freely to evolve during the model run) (Spahni et al., 2013). However, in such case the upscaled $CH_4$ emission estimates would not be any more model independent and therefore less suitable for model validation. We also note that many process models have only one PFT per wetland.

Different variables related to water input to the ecosystem (i.e. P, $P_{ann}$) or surface moisture (SRWI) did not enhance the RF model predictive performance, recognizing that water table depth (WTD) is not solely controlled by input of water via precipitation, but also evapotranspiration and lateral flows affect wetland WTD, data that were missing from our study. These findings are consistent with previous studies (e.g. Christensen et al. 2003, Rinne et al. 2018, Pugh et al. 2018 and Knox et al. (in review)) who showed only a modest $CH_4$ flux dependence on WTD, if any. In contrast, several chamber-based studies have

shown a positive relationship between WTD and $CH_4$ fluxes (Olefeldt et al., 2012; Treat et al., 2018; Turetsky et al., 2014). In general, chamber-based studies often show $CH_4$ flux dependency on WTD whereas studies done at ecosystem scale with EC generally do not, albeit there are exceptions (e.g. Zona et al., 2009). This might indicate that WTD controls meter scale spatial heterogeneity of $CH_4$ flux between microtopographical features (e.g. Granberg et al., 1997) but not temporal variability on the ecosystem scale, provided that WTD stays relative close to the surface. Also, the chamber studies tend to observe spatial

variation, which can be indirectly influenced by WTD via its influence to plant communities, whereas EC studies observe typically temporal variation in sub-annual timescales. However, the effect of WTD might be masked by a contradicting effect caused by plant phenology, since vegetation biomass often peaks at the same time as the WTD is at its lowest. While the variables related to WTD did not increase the RF model performance, WTD might still play a role in controlling ecosystem scale $CH_4$ variability when it is exceptionally high or low. For instance, the year 2006 was exceptionally dry at the Siikaneva

fen and hence $CH_4$ emissions during that year were lower than on average (cf. Fig. 5a). However, in order to accurately capture such dependencies with the machine learning techniques (such as RF), they should be frequent enough so that the model can learn these dependencies.

RF model performance was better at non-permafrost than at permafrost sites and this likely related to the fact that the LSTn is not a good proxy for the temperature controlling the $CH_4$ production and oxidation rates at sites with permafrost. Also,

information about active layer depth was not included here. Furthermore, Zona et al. (2016) showed strong hysteresis between soil temperatures and $CH_4$ emissions at their permafrost sites in Alaska, whereas for instance Rinne et al. (2018) show a synchronous exponential dependence between soil temperature and $CH_4$ emissions at a boreal fen without permafrost. The hysteresis observed in Zona et al. (2016) could be explained by the fact that part of the produced $CH_4$ at these permafrost sites is stored below ground for several months before it is emitted to the atmosphere causing a long temporal lag between soil

temperature and surface flux which would emerge as a hysteresis between soil temperature and $CH_4$ emission. In any case more knowledge on soil processes (soil thawing and freezing, $CH_4$ production and storage) are needed before the $CH_4$ emissions from these permafrost ecosystems can be extrapolated to other areas with greater confidence.

It should be emphasized that the drivers causing across site variability in ecosystem scale $CH_4$ emissions are in general unknown since studies comparing EC $CH_4$ fluxes from multiple wetland sites have only recently been published (Baldocchi,



2014; Knox et al., in review; Petrescu et al., 2015). Most of the past synthesis studies have concentrated on plot-scale measurements (Bartlett and Harriss, 1993; Olefeldt et al., 2012; Treat et al., 2018; Turetsky et al., 2014), however the $CH_4$ flux responses to environmental drivers might be somewhat different at ecosystem scale since $CH_4$ fluxes typically show significant spatial variability on sub-m scale (e.g. Sachs et al., 2010). Furthermore, the temporal coverage of plot-scale measurements

with chambers is usually relatively poor, whereas EC measurements provide continuous data on ecosystem scale. This study and Knox et al. (in review) show that temperature is important when predicting $CH_4$ flux variability in a multisite $CH_4$ flux dataset, but significant fraction of $CH_4$ flux variability is still left unexplained. It remains a challenge for future EC $CH_4$ flux synthesis studies to discover the drivers explaining the rest of the variability.

### 4.2.2 Quality and representativeness of $CH_4$ flux data

The RF model performance may improve if instrumentation, measurement setup and the data processing are harmonized across sites, since these discrepancies between flux sites might have caused spurious differences in $CH_4$ fluxes. These differences would have created additional variability in the whole dataset which would in turn 1) influence the training of RF model and 2) decrease e.g. NSE values obtained against validation data since there would be artificial variability in the validation data which is not related to the predictors. In this study, the site PIs processed the data themselves using different processing codes,

albeit the gapfilling was done centrally in a standardized way.

While these issues mentioned above could impact the upscaling results shown here, prior studies have shown that the usage of different instruments or processing codes do not significantly impact $CH_4$ flux estimates. For instance, Mammarella et al. (2016) showed that the usage of different processing codes (EddyPro and EddyUH) resulted in general in 1 % difference in long-term $CH_4$ emissions. On the other hand, $CH_4$ instrument cross comparisons have shown small differences (typically less

than 7 %) between the long term $CH_4$ emission estimates derived using different instruments (Goodrich et al., 2016; Peltola et al., 2013, 2014). While these studies show consistent $CH_4$ emissions they also stress that the data should be carefully processed to achieve such good agreement across processing codes and instruments. In addition, many issues related to e.g. friction velocity filtering and gapfilling of $CH_4$ fluxes are still unresolved, and the role of short-term emission bursts, which are common in methane flux time series, needs to be further investigated (e.g. Schaller et al., 2017). However, recently Nemitz et

al. (2018) advanced these issues by proposing a methodological protocol for EC measurements of $CH_4$ fluxes used to standardize $CH_4$ flux measurements within the ICOS measurement network (Franz et al, 2018).

Twenty-five flux measurement sites were included in this study and they were distributed across the Arctic-Boreal region (see Fig. 1). The measurements were largely concentrated in the Fennoscandia and Alaska, whereas data from e.g. the HBL and WSL areas were missing. Long-term $CH_4$ flux measurements are largely missing from these vast wetland areas casting

uncertainty on wetland $CH_4$ emissions from these areas. The location of a flux site is typically restricted by practical limitations related to e.g. ease of access and availability of grid power. Hence open-path instruments with low power requirements potentially open up new areas for flux measurements (McDermitt et al., 2010), yet they need continuous maintenance which is not necessarily easy in remote locations. However, one could argue that the geographical location of flux sites is not vital

for upscaling, more important is that the available data represents well the full range of CH$_4$ fluxes across the northern latitudes and more importantly the CH$_4$ flux responses to the environmental drivers. Also, sites should ideally cover all different wetlands with varying species composition, whereas geographical representation is not necessarily as important. CH$_4$ flux site representativeness could be potentially assessed in the same vein as in previous studies for other measurement networks

(Hargrove et al., 2003; Hoffman et al., 2013; Papale et al., 2015; Sulkava et al., 2011). However, before such analysis can be done, the main drivers causing across sites variability in ecosystem scale CH$_4$ fluxes should be better identified.

Most of the CH$_4$ flux data here and in the literature have been recorded during the growing season when the CH$_4$ fluxes are at maximum, whereas year-round continuous CH$_4$ flux measurements are not as common. This is likely due to the harsh conditions in the Arctic during winter which make continuous high-quality flux measurements very demanding (e.g. Goodrich

et al., 2016; Kittler et al., 2017a), but also in part since the large-scale importance of nongrowing season emissions has just recently been recognized (Kittler et al., 2017b; Treat et al., 2018; Xu et al., 2016a; Zona et al., 2016). For upscaling year-round CH$_4$ emissions, continuous measurements are vital to accurately constrain also the nongrowing season emissions and their drivers.

## 5 Data availability

The presented upscaled CH$_4$ flux maps (RF-DYPTOP, RF-PEATMAP and RF-GLWD) and their uncertainties are accessible via an open-data repository Zenodo (Peltola et al., 2019). The datasets are saved in netCDF-files and they are accompanied by a read me file. The dataset can be downloaded from https://doi.org/ 10.5281/zenodo.2560164.

## 6 Conclusions

Methane (CH$_4$) emission data consisting of over 40 site-years from 25 eddy covariance flux measurement sites across the

Arctic-Boreal region were assembled and upscaled to estimate CH$_4$ emissions from northern (>45 °N) wetlands. The upscaling was done using the random forest (RF) algorithm. The performance of the RF model was evaluated against independent validation data utilizing the leave-one-site-out scheme which yielded value of 0.47 for both the Nash-Sutcliffe model efficiency and R$^2$. These results are similar to previous upscaling studies for the net ecosystem exchange of carbon dioxide (NEE) but are less good than for the individual components of NEE or energy fluxes (e.g. Jung et al., 2010; Tramontana et al., 2016). The

performance is also comparable to studies where process models are compared against site CH$_4$ flux measurements (McNorton et al., 2016; Wania et al., 2010; Zürcher et al., 2013; Zhu et al., 2014; Xu et al., 2016a). Hence, despite the relatively high fraction of unexplained variability in the CH$_4$ flux data, the upscaling results are useful for comparing against models and could be used to evaluate model results. The three gridded CH$_4$ wetland flux estimates and their uncertainties are openly available for further usage (Peltola et al., 2019).



The upscaling to the regions > 45 °N resulted in mean annual CH$_4$ emissions comparable to prior studies on wetland CH$_4$ emissions from these areas (Bruhwiler et al., 2014; Chen et al., 2015; Spahni et al., 2011; Treat et al., 2018; Watts et al., 2014; Zhang et al., 2016; Zhu et al., 2013) and hence in general support the prior modelling results for the northern wetland CH$_4$ emissions. When compared to two validation areas, the upscaling likely overestimated CH$_4$ emissions from the Hudson Bay

5    lowlands, whereas emission estimates for the western Siberian lowlands were in a reasonable range. Future CH$_4$ flux upscaling studies would benefit from long-term continuous CH$_4$ flux measurements, centralized data processing and better incorporation of CH$_4$ flux drivers (e.g. wetland vegetation composition and carbon cycle) from remote sensing data needed for scaling the fluxes from the site level to the whole Arctic-boreal region.

## Appendix A

10   **Table A1. Description of eddy covariance sites included in this study.**

| Name | Site ID | Site PI | Latitude, Longitude | Amount of monthly CH$_4$ flux data available | Reference | sedges as dominant vegetation type (true/false) | permafrost present (true/false) | Biome based on Olson et al. (2011) | Wetland type | Time resolution of data |
|---|---|---|---|---|---|---|---|---|---|---|
| Schechenfilz Nord | DE-SfN | Janina Klatt, Hans Peter Schmid | 47.8064, 11.3275 | 2 | Hommeltenberg et al. (2014) | false | false | temperate | bog | 30 min |
| Chokurdakh | RU-Cok | Albertus J. Dolman | 70.8291, 147.4943 | 5 | Parmentier et al. (2011) | true | true | tundra | wet tundra | 30 min |
| Vorkuta | RU-Vor | Thomas Friborg | 67.0547, 62.9405 | 5 | Marushchak et al. (2016) | false | true | tundra | wet tundra | 30 min |
| Stordalen | SE-Stl | Thomas Friborg | 68.3542, 19.0503 | 6 | Jammet et al. (2017) | true | false | tundra | fen | 30 min |





| | | | | | | | | | | |
|---|---|---|---|---|---|---|---|---|---|---|
| Stordalen (ICOS)* | SE-Sto | Janne Rinne | 68.3560, 19.0452 | 55 | | false | true and false[a] | tundra | bog | 30 min |
| Siikaneva 1 | FI-Sii | Timo Vesala, Ivan Mammarella | 61.8327, 24.1928 | 104 | Rinne et al. (2018) | true | false | boreal | fen | 30 min |
| Siikaneva 2 | FI-Si2 | Timo Vesala, Ivan Mammarella | 61.8375, 24.1699 | 26 | Korrensalo et al. (2018) | false | false | boreal | bog | 30 min |
| Lompolojänkkä | FI-Lom | Annalea Lohila | 67.9972, 24.2092 | 59 | Aurela et al. (2009) | true | false | boreal | fen | 30 min |
| James Bay lowlands | CA-JBL | Daniel F. Nadeau | 53.6744, -78.1706 | 3 | Nadeau et al. (2013) | false | false | boreal | bog | daily |
| Lost Creek | US-Los | Ankur R. Desai | 46.0827, -89.9792 | 30 | Pugh et al. (2018) | false | false | temperate | fen | 30 min |
| Atqasuk | US-Atq | Donatella Zona | 70.4696, -157.4089 | 11 | Zona et al. (2016) | true | true | tundra | wet tundra | 30 min |
| Barrow Environmental Observatory | US-Beo | Donatella Zona | 71.2810, -156.6123 | 16 | Zona et al. (2016) | true | true | tundra | wet tundra | 30 min |





| | | | | | | | | | | |
|---|---|---|---|---|---|---|---|---|---|---|
| Biocomplexity Experiment South tower | US-Bes | Donatella Zona | Zona et al. (2016) | 16 | 71.2809, -156.5965 | true | true | tundra | wet tundra | 30 min |
| Ivotuk | US-Ivo | Donatella Zona | Zona et al. (2016) | 15 | 68.4865, -155.7502 | true | true | tundra | wet tundra | 30 min |
| Western peatland 1 | CA-WP1 | Lawrence B. Flanagan | Long et al. (2010) | 5 | 54.9538, -112.4670 | false | false | temperate | fen | 30 min |
| Mer Bleue | CA-Mer | Elyn Humphreys | Brown et al. (2014) | 16 | 45.4094, -75.5186 | false | false | temperate | bog | daily |
| Chersky reference | RU-Ch2 | Mathias Göckede | Kittler et al. (2017) | 21 | 68.6169, 161.3509 | true | true | boreal | wet tundra | daily |
| Rzecin | PL-wet | Bogdan Chojnicki | Kowalska et al. (2013) | 4 | 52.7622, 16.3094 | true | false | temperate | fen | 30 min |
| Degerö Stormyr | SE-Deg | Mats B. Nilsson | Nilsson et al. (2008) | 22 | 64.1820, 19.5567 | true | false | boreal | fen | 30 min |
| Seney | US-Sen | Thomas Pypker | Pypker et al. (2013) | 5 | 46.3167, -86.0500 | true | false | temperate | fen | daily |





| Scotty Creek | CA-SCC | Oliver Sonnentag | 61.3000, -121.3000 | 14 | Helbig et al. (2016) | false | false | boreal | bog | 30 min |
| Samoylov | RU-Sam | Torsten Sachs | 72.3667, 126.5000 | 11 | Sachs et al. (2008) | true | true | tundra | wet tundra | 30 min |
| Innavait Creek | US-ICh | Eugenie S. Euskirchen | 68.6060, -149.3110 | 7 | | true | true | tundra | wet tundra | 30 min |
| Bonanza Creek, fen | US-BCF | Eugenie S. Euskirchen | 64.7040, -1483130 | 16 | Euskirchen et al. (2014) | true | false | boreal | fen | 30 min |
| Bonanza Creek, bog | US-BCB | Eugenie S. Euskirchen | 64.7000, -148.3200 | 14 | Euskirchen et al. (2014) | false | false | boreal | bog | 30 min |

[a] Data from this site is divided into two since data from two wind directions differ from each other (with and without permafrost).

**Author contribution**

OP, TA and TV designed the study and YG contributed further ideas for the study. OP did the data processing and analysis, except OR prepared the PEATMAP map for the study. PA, MA, BC, ARD, AJD, ESE, TF, MG, MH, EH, GJ, JK, LK, AL, IM, DFN, MBN, WCO, MP, TP, WQ, JR, TS, MS, HPS, OS, CW and DZ provided $CH_4$ fluxes and other in-situ data for the study. FJ and SL did the LPX-Bern model runs. OP wrote the first version of the manuscript and all the authors commented it and made modifications.



**Acknowledgments**

Lawrence B. Flanagan is acknowledged for providing data from CA-WP1 site. Lawrence B. Flanagan acknowledge support from the Natural Sciences and Engineering Council of Canada and Canadian Foundation for Climate and Atmospheric Sciences. OP is supported by the Postdoctoral Researcher project (decision 315424) funded by the Academy of Finland. OR is supported by the Academy of Finland IIDA-MARI project (decision 313828). The financial support by the Academy of Finland Centre of Excellence (272041 and 307331), Academy Professor projects (312571 and 282842), ICOS-Finland (281255), CARB-ARC project (285630) are acknowledged. SHK and RBJ acknowledge support from the Gordon and Betty Moore Foundation through Grant GBMF5439 'Advancing Understanding of the Global Methane Cycle'. ARD acknowledges support of the DOE Ameriflux Network Management Project. AJD acknowledges support from the Netherlands Earth System Science Centre, NESSC). TS was supported by the Helmholtz Association of German Research Centres (grant VH-NG-821). IM and TV thank the EU for supporting the RINGO project funded by the Horizon 2020 Research and Innovation Programme (grant agreement 730944). Also EU-H2020 CRESCENDO project (641816) is acknowledged.

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





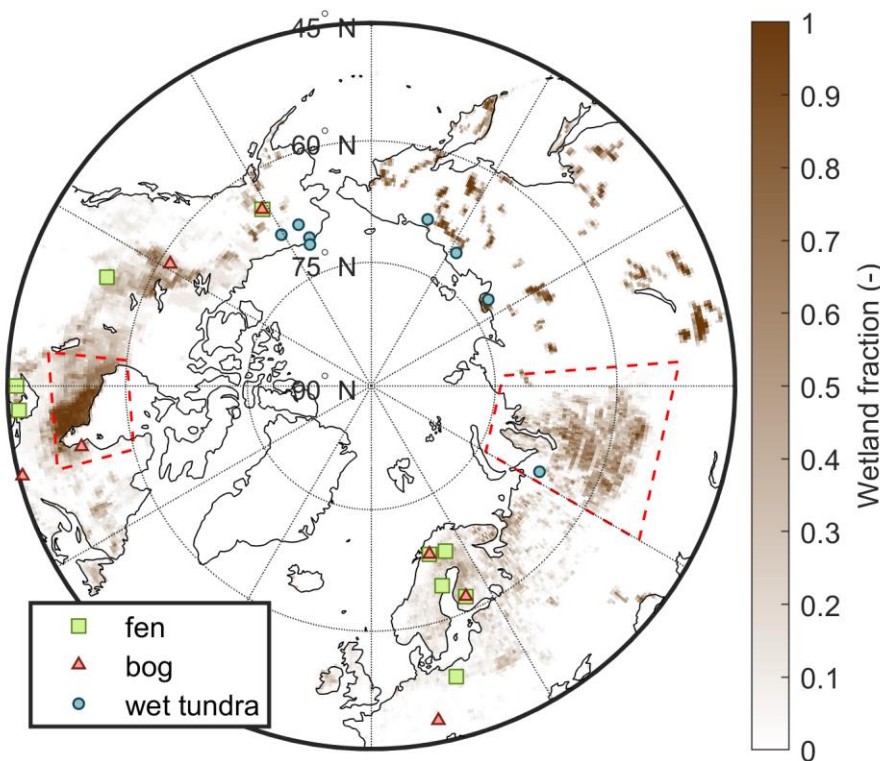

**Figure 1: Map showing the locations of the EC measurements. The distribution of wetlands shown in the figure is based on Xu et al. (2018). Hudson Bay lowlands (50°N-60°N, 75°W-96°W) and western Siberian lowlands (52°N-74°N, 60°E-94.5°E) are highlighted with red dashed lines.**





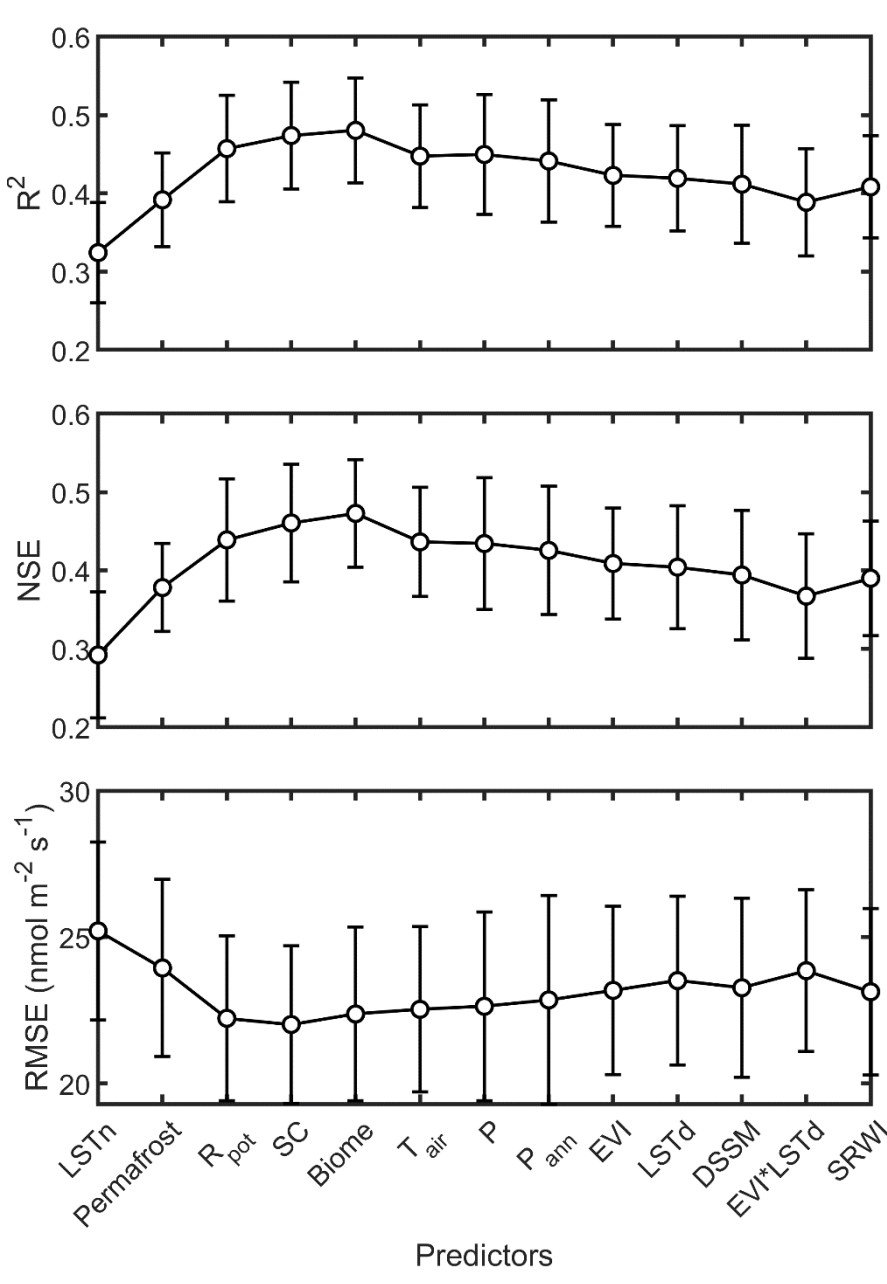

**Figure 2. Evolution of statistical metrics during RF model development. Predictors were added to the RF model starting from the left of the figure and accumulate along the x-axis. For instance, the x-tick label "SC" shows the RF model performance when LSTn, Permafrost, Rpot, and SC were used as predictors in the model. See the x-tick label explanations in Table 1. The error bars denote 1-sigma uncertainty of the values estimated with bootstrapping.**





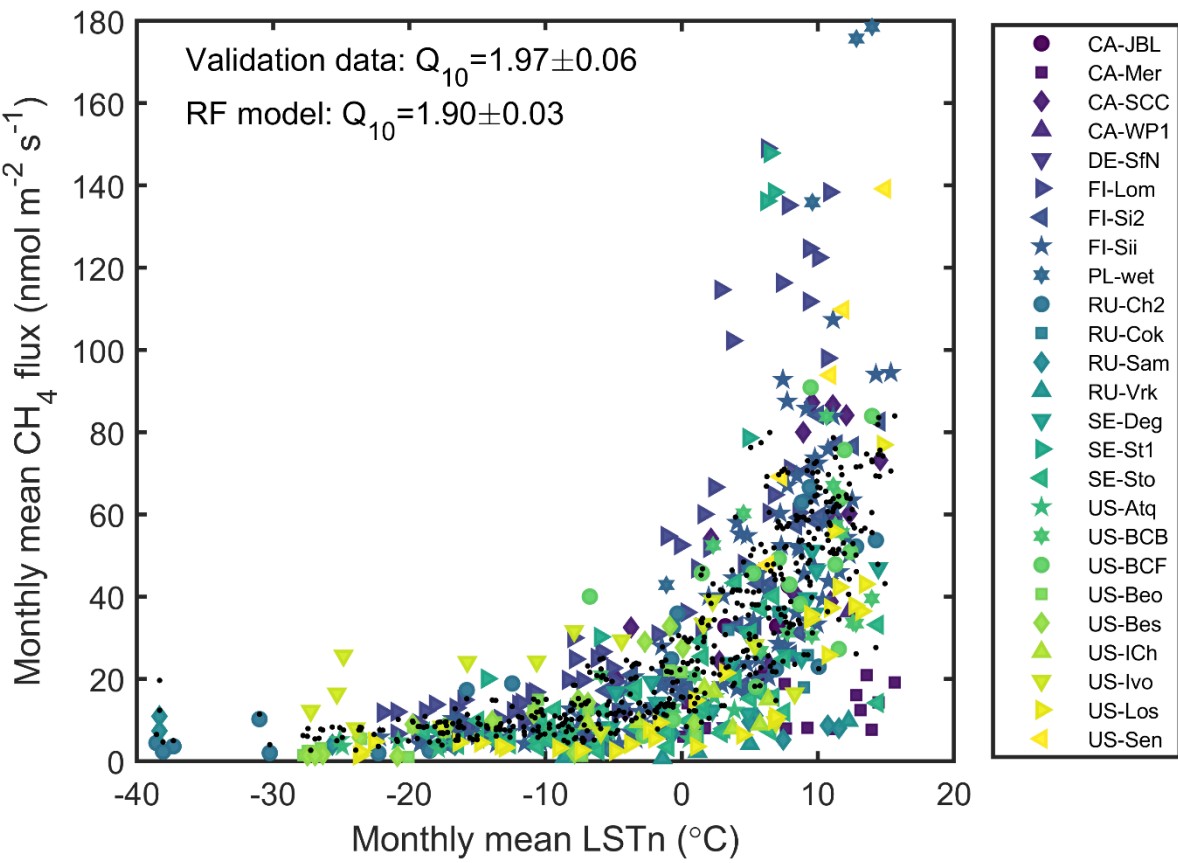

**Figure 3. Dependence of monthly mean CH₄ emissions on monthly mean land surface temperature at night (LSTn) derived from MODIS data. EC measurements are shown with filled markers (unique colour for each site) and RF model predictions for each site are given with black dots.**





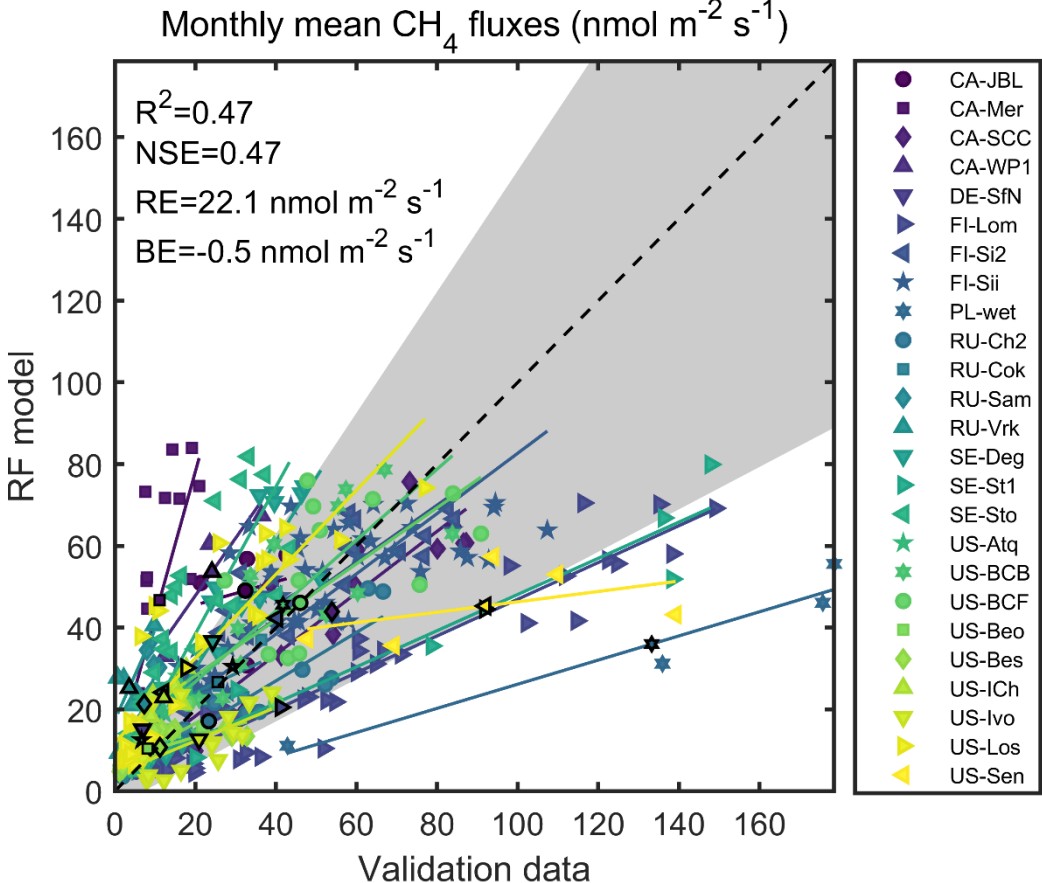

**Figure 4. Relation between monthly mean CH₄ fluxes predicted by the RF model and independent validation data. Monthly average values from the same site are identified by unique colours and least squares linear fit to data from each site is also plotted using the same colour. Site means are shown with markers with black edges. The dashed line shows the 1:1 line. The shaded area shows the uncertainty range estimated from the RE CH₄ flux dependence (see text for further details). The statistics in the figure are calculated using the monthly data.**



**Figure 5.** Time series of modelled CH₄ emissions (red lines) together with validation data (circles) at four example sites: a) Siikaneva oligotrophic fen in Finland, b) Lost Creek shrub fen in Wisconsin, US, c) Atqasuk wet tundra in Alaska, US and d) Chersky wet tundra in northeast Siberia, Russia. Vertical dashed lines denote a new year. Note the changes in y-axis scales. Site specific model performance metrics are also included.




**Figure 6. Mean annual CH$_4$ wetland emissions during years 2013-2014 estimated by upscaling EC data using the RF model and three wetland maps (top row) and process models (bottom row). Grid cells with low CH$_4$ wetland emissions (below 0.1 g(CH$_4$) m$^{-2}$ year$^{-1}$) are shown with grey. The flux rates refer to total unit area in a grid cell.**





**Figure 7. Difference in mean annual CH₄ wetland emissions during years 2013-2014 estimated by upscaling EC data using the RF model with different wetland maps and process models. All the CH₄ emission maps were aggregated to 1° resolution before comparison. The flux rates refer to total unit area in a grid cell.**




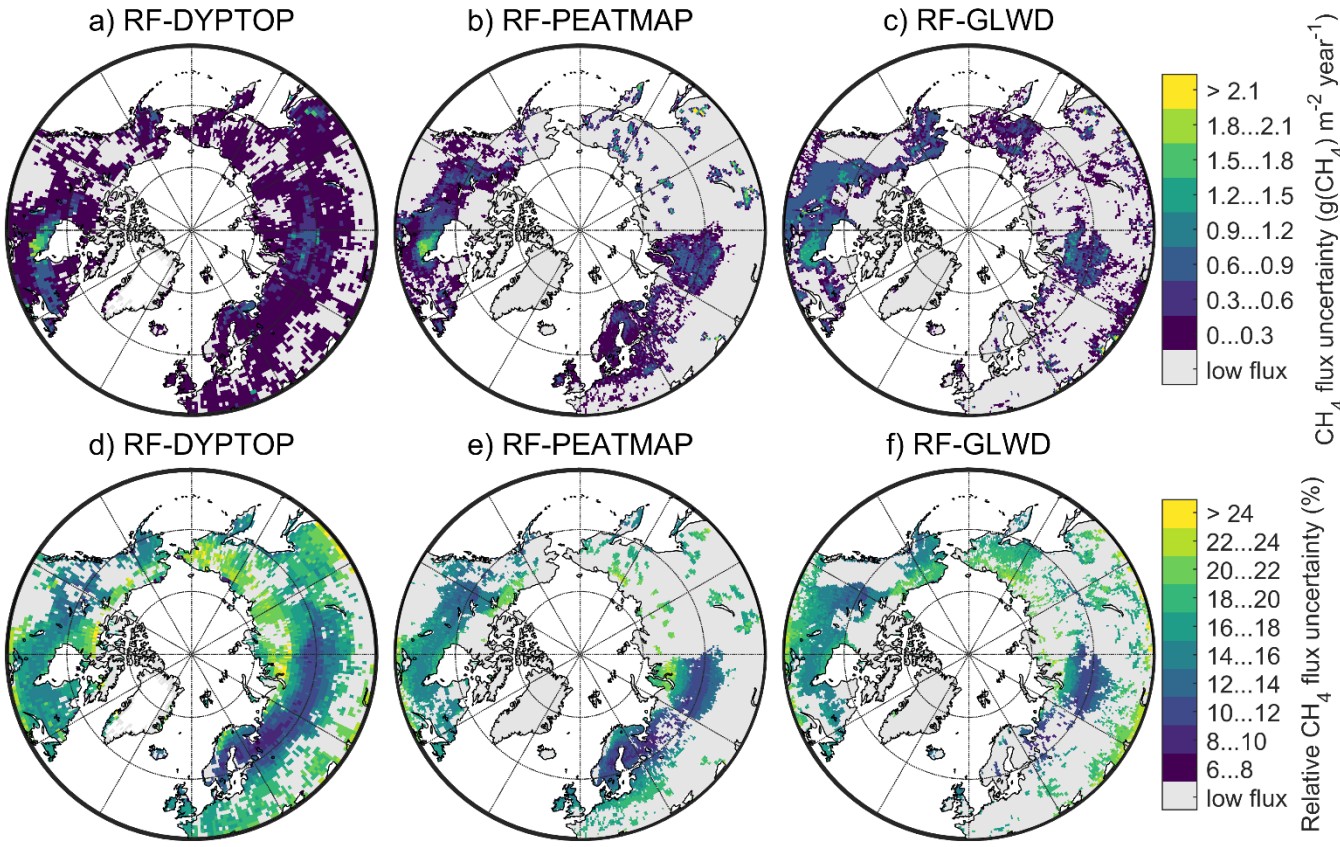

**Figure 8. Absolute (subplots a)-c)) and relative (subplots d)-f)) uncertainties of the upscaled CH₄ fluxes using different wetland maps. Uncertainty is estimated as 1-σ variability of the predictions by 200 RF models developed by bootstrapping the training data (Sect. 2.1.2). Grid cells with low CH₄ wetland emissions (below 0.1 g(CH₄) m⁻² year⁻¹) are shown with grey. The absolute uncertainties refer to total unit area in a grid cell.**





**Figure 9. Monthly time series of zonal mean CH₄ fluxes. The upscaled fluxes with different wetland maps are shown in subplots a), b) and c) and wetland CH₄ emissions estimated with the two process models are given in subplots d) and e).**



**Table 1. Description of input variables for RF model development for upscaling. Data were aggregated to monthly values (see text) unless otherwise noted below.**

| | Name | Description | Data source | Available in gridded format |
|---|---|---|---|---|
| Site measurements | $T_{air}$ | Mean air temperature | site PI & WFDEI | Yes |
| | P | Precipitation | site PI & WFDEI | Yes |
| | $P_{ann}$ | Annual precipitation | site PI & WFDEI | Yes |
| Remote sensing | LSTn | Land surface temperature at night | MOD11A2 | Yes |
| | LSTd | Land surface temperature at day | MOD11A2 | Yes |
| | EVI | Enhanced vegetation index | MOD13A3 | Yes |
| | SRWI | Simple ratio water index (SRWI = $R_{858}/R_{1240}$) | MOD09A1 | Yes |
| | SC | Snow cover flag | MOD10A1 | Yes |
| | EVI* LSTd | Product of EVI and LSTd, a proxy for GPP (Schubert et al., 2010) | MOD13A3 & MOD11A2 | Yes |
| Additional categorical variables | Permafrost | Flag for permafrost at site (true/false) | site PI | Yes |
| | Biome | Site classification based on biome (temperate, boreal and tundra) | Olson et al. (2001) | Yes |
| | type | Wetland type (fen, bog, tundra) | site PI | No |
| | sedge | flag for sedges as dominant vegetation type (true/false) | site PI | No |
| Other | $R_{pot}$ & der($R_{pot}$) | Potential solar radiation at the top of atmosphere and its first time derivative | - | Yes |
| | DSSM | Days since snowmelt, derived from the snow cover flag | - | Yes |



**Table 2. Annual CH₄ wetland emissions in different subdomains (Hudson Bay lowlands and Western Siberian lowlands, see Fig. 1) and time periods. The values are given in Tg(CH₄) year⁻¹. Note that estimates from some reference studies are not for the same period as the one studied here (2013-2014). For WetCHARTs the mean of the model ensemble together with the range (in parentheses) are given, whereas for the upscaling results the 95 % confidence intervals for the estimated emissions are given.**

|  | Reference | Hudson Bay lowlands | Western Siberian lowlands | Nongrowing season fluxes from northern wetlands (November…March) | Annual emissions north from 45 °N |
|---|---|---|---|---|---|
| Inversion models | Bohn et al. (2015), WETCHIMP-WSL |  | 6.06 ± 1.22 |  |  |
|  | Bruhwiler et al. (2014)[a] |  |  |  | 23 |
|  | Kim et al. (2011) |  | 2.9 ± 1.7 and 3.0 ± 1.4 |  |  |
|  | Miller et al. (2014) | 2.4 ± 0.3 |  |  |  |
|  | Spahni et al. (2011) |  |  |  | 28.2 ± 2.2 |
|  | Thompson et al. (2017) | 2.7-3.4 | 6.9 ± 3.6 |  |  |
| Process models | Bohn et al. (2015), WETCHIMP-WSL |  | 5.34 ± 0.54 |  |  |
|  | Chen et al. (2015)[b] | 3.11 ± 0.45 |  |  | 35.0 ± 6.7 |
|  | Melton et al. (2013), WETCHIMP[c] | 5.4 ± 3.2 |  |  |  |
|  | Pickett-Heaps et al. (2011)[d] | 2.3 ± 0.3 |  |  |  |
|  | Treat et al. (2018)[e] |  |  | 6.1 ± 1.5 | 37 ± 7 |
|  | Watts et al. (2014) |  |  |  | 53 |
|  | Zhang et al. (2016)[f] | 5.5 ± 1.1 | 4.6 ± 0.6 |  | 30.3± 5.4 |
|  | This study, LPX-Bern | 2.5 | 4.4 | 4.5 | 24.7 |
|  | This study, WetCHARTs | 2.8 (0.5-8.7) | 4.2 (1.6-9.4) | 5.1 (0.6-17.0) | 29.7 (8.7-74.0) |
| Flux measurement upscaling | Glagolev et al. (2011) |  | 3.9 ± 1.3 |  |  |
|  | Zhu et al. (2013) |  |  |  | 44.0-53.7 |
|  | This study, RF-PEATMAP | 4.8 (3.3-6.3) | 6.6 (4.9-8.4) | 6.7 (4.9-8.5) | 31.7 (22.3-41.2) |



| | | | | | |
|---|---|---|---|---|---|
| | This study, RF-DYPTOP | 4.6 (3.1-6.0) | 7.0 (5.2-8.8) | 6.2 (4.6-7.8) | 30.6 (21.4-39.9) |
| | This study, RF-GLWD | 4.9 (3.4-6.5) | 6.8 (5.0-8.5) | 8.0 (5.8-10.2) | 37.6 (25.9-49.5) |

a Approximately north from 47 °N

b Approximately north from 45 °N

c Mean annual CH$_4$ emissions from eight models ± 1-sigma of interannual variation in the model estimates for the period 1993-2004.

5    d Process model tuned to match atmospheric observations

e North from 40 °N

f Mean ± 1-sigma over LPJ-wsl model results using different wetland extends for the period 1980-2000.