# Peer review of "Monthly Gridded Data Product of Northern Wetland Methane"

_Earth System Science Data, 2019_

## Referee Comment (RC1) · Anonymous Referee #1 · 11 Apr 2019

The authors use a machine learning technique (random forest-RF) to develop a monthly gridded data product of northern (>= 45°C) wetland CH4 emissions. Three CH4 emissions products (2013-2014) are derived from RF based on different wetland maps. Annual total CH4 emissions from these three products are comparable to previous studies and two process-based models, however, the areal extent and spatial-temporal patterns of the CH4 emissions are largely subject to the wetland map. These products can potentially become a good benchmark for both top-down and bottom-up models. Overall, the manuscript is well structured and written. The methods and results follow well the objectives. The figures clearly illustrate the results. However, I think there are still some aspects this study should spend more efforts to justify before

the manuscript can be accepted in the final publication of Earth System Science Data:

(1) Evaluation of the products is too simplistic. It would be nice to illustrate the statistics for model-observation comparison spatially, which means to make a plot to see the spatial patterns of R2, NSE, RE and BE.

There are two hot spots for CH4 emission across the circumpolar region (Hudson Bay lowlands and western Siberian lowlands). But unfortunately, there is only one site from each hot spot area. Why don't the authors select these two sites for the time series comparison, like what is shown in Figure 5? I notice that Figure 4 indicates regression lines between model and observations for these two sites have large deviations from the 1:1 line.

(2) Why does the study only select two years of observations? As most observations in the Arctic have only been collected during growing seasons, there are not many monthly data points left within two years. Therefore, it is difficult to say how significant the regression relationships between model and observation in each site is.

(3) Why do the authors compare RF-GLWD with LPJ-Bern and WetCHARTs? I don't see the point of making such a comparison, because they are not based on the same wetland map.

(4) I noticed the input variables listed in table 1 comprise of some categorical variables. Can the authors describe more details about how RF use such information? Can the authors consider making an RF model based on individual wetland type or biome type? Moreover, why don't these input variables include water table position, which is important drivers for CH4 emissions, particularly to explain spatial heterogeneity?

Specific comments:

Line 11: utilize random forest (RF) -> utilize a random forest (RF)

Line 16: What does "confidence interval" refer to? a random forest ensemble? Please specify it.

Line 27: constraint -> constrain

---

## Referee Comment (RC2) · Anonymous Referee #2 · 17 Apr 2019

The authors have produced a gridded product of monthly CH4 fluxes from Northern (>=45°) wetlands using a random forest (machine learning) algorithm. The random forest model was trained against a number of predictor variables such as nighttime Land Surface Temperature, absence or presence of permafrost, potential sunshine, sedge cover and biome class. Eddy covariance measurements of CH4 from 25 sites was used to optimize model performance during the training. The fluxes were then upscaled using three different wetland maps and compared against previous studies and two process models. There results are comparable to these.

The manuscript is well written and explanatory, highlights uncertainties and discuss

data in a good way. The product have potential to become valuable to process modellers as a benchmarking product further on, which is also the aim of the study.

I have a few thoughts that could be considered before the final report is submitted.

1) The most important predictor variable is the nighttime land surface temperature (LSTn). While this is interesting and maybe not completely unexpected I lack the discussion around this. Is there a mechanism behind the nighttime temperature or is it an outcome of the machine learning algorithm? For instance, I would assume that most process based models would use daily temperature as that would be better correlated with ecosystem productivity.

2) In the upscaling process you initially calculate the flux density per grid cell and then use the wetland maps to produce the amount of CH4 per grid cell. As the maps used contribute highly to the uncertainties in these amounts I would consider to add a map of the flux densities. Such a map could also be used as a benchmark product by bottom-up models.

3) The comparison to the process models contains uncertainty both in the wetland map used as pointed out by referee 1, but also in the climate forcing data of the models. This should maybe at least be clarified if a comparison should be made.

4) In the introduction the contribution of CH4 emissions from natural wetland is discussed (page 2, line 30-33). Although it might be implicitly understood, I lack a clarification that these are global estimates.

Technichal corrections: 1) I cannot find where the abbreviation of SC is defined. I assume that it stands for sedge cover?

2) Page 14, line 11 th an -> than

3) Page 14 line 12. Consider not using . . . for timespans. Use instead, e.g., years 2008 - 2011

4) Page 14, line 15, same as 3)

5) Also why not use subscript for ecosystem respiration Reco -> $R_{eco}$?

---

## Author Comment (AC1) · 16 Jun 2019

Please find our replies and revised manuscript in the supplement.

On behalf of the authors, Olli Peltola

Please also note the supplement to this comment:
https://www.earth-syst-sci-data-discuss.net/essd-2019-28/essd-2019-28-AC1-supplement.zip

2019.

---

## Author Response (AR1)

AUTHOR RESPONSES TO REVIEWER COMMENTS

Title: Monthly Gridded Data Product of Northern Wetland Methane Emissions Based on Upscaling Eddy Covariance Observations

MS No.: essd-2019-28

We thank the reviewers for their comments on our manuscript. The comments helped us to identify the weaknesses and unclear sections in our manuscript and hence helped us to improve the text. We address the individual comments point-by-point below. The comments are with **bold** font whereas the responses to the comments are with normal font.

Reviewer #1

**(1) Evaluation of the products is too simplistic. It would be nice to illustrate the statistics for model-observation comparison spatially, which means to make a plot to see the spatial patterns of R2, NSE, RE and BE.**
**There are two hot spots for CH4 emission across the circumpolar region (Hudson Bay lowlands and western Siberian lowlands). But unfortunately, there is only one site from each hot spot area. Why don't the authors select these two sites for the time series comparison, like what is shown in Figure 5? I notice that Figure 4 indicates regression lines between model and observations for these two sites have large deviations from the 1:1 line.**

RESPONSE: Good idea. We will add a figure showing the spatial patterns of the suggested statistical metrics (i.e. R2, NSE, RE and BE). However, one should note that BE is already shown in Fig. 7 of the manuscript. Also, in order to avoid effects of differences in wetland maps, we will add only the comparison between LPX-Bern and RF-DYPTOP since they are based on the same wetland map.

The second point the reviewer raises in the comment is also most valid. It would make sense to select the sites from Hudson Bay lowlands (HBL) and western Siberian lowlands (WSL) (CA-JBL and RU-Vrk, respectively) for time series comparison. However, we opted not to do that since we have only limited amount of monthly data points from these two sites (four and five data points, respectively) and hence time series plots would not be very informative. Hence, we opt not to add these sites to Fig. 5 of the manuscript.

CHANGES IN MANUSCRIPT: added a figure (Fig.8 in the revised manuscript) showing maps of the statistical metrics (NSE, R2 and RE) comparing LPX-Bern and RF-DYPTOP.

**(2) Why does the study only select two years of observations? As most observations in the Arctic have only been collected during growing seasons, there are not many monthly data points left within two years. Therefore, it is difficult to say how significant the regression relationships between model and observation in each site is.**

RESPONSE: We used all the available observations for model development and validation, not only 2013-2014. The upscaling work flow was as follows: 1) we gathered all available CH4 flux eddy covariance observations from northern wetlands, 2) developed random forest (RF) machine learning model utilizing all this data, 3) used the RF model and spatial maps of CH4 flux drivers to upscale the measurements for two years (2013 and 2014). This can be seen in the multi-year time series figure 5 where RF model predictions are validated against all the available data from four EC sites. Also, the main RF model validation figure (Fig. 4) contains all the available observations from the years 2005-2016, not only 2013-2014.

For upscaling, we selected only the two years 2013 and 2014 mainly since in these two years, the largest overlapping fraction of the available observations were made.

CHANGES IN MANUSCRIPT: In order to clarify this point we will emphasise in the abstract, Sect. 2.3.1 and Sect. 4.2.1 that all the available data from all the years were used in the model development and validation.

**(3) Why do the authors compare RF-GLWD with LPJ-Bern and WetCHARTs? I don't see the point of making such a comparison, because they are not based on the same wetland map.**

RESPONSE: We partly agree with the reviewer that direct point-by-point comparison of products not based on the same wetland map adds complexity. Yet, here, we wanted to emphasise the fact that the underlying wetland maps have a big impact on the estimated spatial distribution of wetland $CH_4$ emissions. Furthermore, it is unclear which wetland map is most accurate for the study domain and hence upscaling with as many wetland maps as feasible is warranted. Hence, we opt to keep the comparison as it is.

CHANGES IN MANUSCRIPT: We will add the following sentence to the first part of Sect. 3.3.: " While the wetland maps differ, there is no consensus on which is more accurate, so comparisons indicate the uncertainty in upscaling emanating from uncertainties in wetland distribution.".

**(4) I noticed the input variables listed in table 1 comprise of some categorical variables. Can the authors describe more details about how RF use such information? Can the authors consider making an RF model based on individual wetland type or biome type? Moreover, why don't these input variables include water table position, which is important drivers for CH4 emissions, particularly to explain spatial heterogeneity?**

RESPONSE: During the training of a RF model the mean squared error (MSE) between training data and RF model output is minimized. The categorical variables are used in the split nodes to divide the data into two in such a way that the MSE is minimized after the split. Let us take a simple example: we develop a RF model with three input variables, snow cover (SC), wetland type (WT) and air temperature (Ta). SC and WT are categorical variables, whereas Ta is a continuous variable. After training, a single decision tree for regressions in a RF model could look like in Fig. 1 below. The output from this one tree would be the mean of the training data in each leaf node. The output from RF model is then an average of the output from all the trees in the RF model.

[Figure]

*Figure 1. Example of a decision tree for regression with three input variables: SC, WT and Ta. SC and WT are categorical variables, whereas Ta is a continuous variable. The output from this particular regression tree is the mean of the training data in each leaf node.*

In principle one could make an RF model on individual wetland or biome type. Yet, in our study, the categorical variables, wetland or biome type, did not increase the overall performance of the developed model and hence they were not included as drivers in the model used in upscaling.

We agree that the water table depth (WTD) is likely an important driver for the wetland CH4 emissions, though a growing body of literature shows temperature to dominate for spatial variation and short-term variation. This is already discussed in the manuscript (Sect. 4.2.1). However, WTD measurements were not available in many of the study sites preventing us from using it for our upscaling procedure.

CHANGES IN MANUSCRIPT: add text in Sect. 2.1 on how RF model uses different input variables (categorical, continuous).

**Specific comments:**
**Page 2**
**Line 11: utilize random forest (RF) -> utilize a random forest (RF)**
RESPONSE: thanks, will be corrected.

**Line 16: What does "confidence interval" refer to? a random forest ensemble? Please specify it.**
RESPONSE: Yes, it refers to 95 % confidence interval of the random forest ensemble. We will specify this in the text.

**Line 27: constraint -> constrain**
RESPONSE: Thanks, will be corrected.

Reviewer #2

**The authors have produced a gridded product of monthly CH4 fluxes from Northern (>=45) wetlands using a random forest (machine learning) algorithm. The random forest model was trained against a number of predictor variables such as nighttime Land Surface Temperature, absence or presence of permafrost, potential sunshine, sedge cover and biome class. Eddy covariance measurements of CH4 from 25 sites was used to optimize model performance during the training. The fluxes were then upscaled using three different wetland maps and compared against previous studies and two process models. There results are comparable to these. The manuscript is well written and explanatory, highlights uncertainties and discuss data in a good way. The product have potential to become valuable to process modellers as a benchmarking product further on, which is also the aim of the study. I have a few thoughts that could be considered before the final report is submitted.**

RESPONSE: Thanks, we acknowledge the positive feedback.

**1) The most important predictor variable is the nighttime land surface temperature (LSTn). While this is interesting and maybe not completely unexpected I lack the discussion around this. Is there a mechanism behind the nighttime temperature or is it an outcome of the machine learning algorithm? For instance, I would assume that most process based models would use daily temperature as that would be better correlated with ecosystem productivity.**

RESPONSE: Most likely it is just an outcome of the machine learning algorithm and the available data. The results using other temperature variables (e.g. daytime land surface temperature (LSTd) or daily mean air temperature (Ta)) as drivers instead of LSTn were almost as good as with LSTn when evaluated against the independent validation data. Possibly with slightly different data set (more sites) other temperature variables (e.g. Ta) might have been more important drivers for the CH4 flux variability. However, we opted to use LSTn since this variable was selected by the driver selection procedure used in this study (Sect. 3.1).

However, it might also be that LSTn is a better proxy of the temperature controlling the CH4 production and oxidation processes in the soil (and hence emissions), since it is less affected by the daytime solar heating than LSTd and Ta. Hence it might better reflect the more dampened temperatures deeper in the soil than the more variable LSTd or Ta. However, this is left for future studies to clarify.

CHANGES IN MANUSCRIPT: Shortly mention in Sect. 3.1. that the selection of LSTn as a primary driver instead of e.g. daily mean air temperature was most likely an outcome of the algorithm and data used in this study.

**2) In the upscaling process you initially calculate the flux density per grid cell and then use the wetland maps to produce the amount of CH4 per grid cell. As the maps used contribute highly to the uncertainties in these amounts I would consider to add a map of the flux densities. Such a map could also be used as a benchmark product by bottom-up models.**

RESPONSE: Yes, we agree a map of flux densities would be a valuable addition to the published data set. Initially, we did not include such map since it is difficult to judge for which grid cells the flux densities would be included in the published data set. Of course, RF models can be used to calculate CH4 flux densities for every grid cell on the globe, but are not very informative for grid cells with

zero wetland coverage. Hence we opted to show only maps where flux densities are scaled by wetland coverage to produce the CH4 emitted per grid cell.

We will add a map of flux densities to the published data set, but limit the data set only to grid cells with significant (>5% grid cell coverage) wetland coverage based on the three wetland maps available.

CHANGES IN MANUSCRIPT: No changes to the manuscript, but we will add wetland CH4 flux densities to the published data set.

**3) The comparison to the process models contains uncertainty both in the wetland map used as pointed out by referee 1, but also in the climate forcing data of the models. This should maybe at least be clarified if a comparison should be made.**

RESPONSE: Good point, we will clarify that there are several sources of uncertainty when comparing process models and the upscaled products, one of them being the different climate forcing data used in the models.

CHANGES IN MANUSCRIPT: Add short text on the different sources of uncertainty when the upscaled products are compared against process models (Sect. 3.3).

**4) In the introduction the contribution of CH4 emissions from natural wetland is discussed (page 2, line 30-33). Although it might be implicitly understood, I lack a clarification that these are global estimates.**

RESPONSE: Thanks, we will clarify that the first part of the introduction discusses global wetland emissions.

CHANGES IN MANUSCRIPT: Clarify in the Introduction that the emission estimates from Saunois et al. (2016) are global estimates.

**Technical corrections: 1) I cannot find where the abbreviation of SC is defined. I assume that it stands for sedge cover?**

RESPONSE: SC stands for snow cover flag. We will define the abbreviation in Sect. 2.3.2 where we introduce the used satellite products.

**2) Page 14, line 11 th an -> than**

RESPONSE: Thanks, will be corrected.

**3) Page 14 line 12. Consider not using … for timespans. Use instead, e.g., years 2008-2011**

RESPONSE: Thanks, will be corrected

**4) Page 14, line 15, same as 3)**

RESPONSE: See our response above.

**5) Also why not use subscript for ecosystem respiration Reco -> $R_{eco}$?**

RESPONSE: Thanks we will change Reco to $R_{eco}$ everywhere in the text.

[revised manuscript text omitted]